# Long-term incubations provide insight into the mechanisms of anaerobic oxidation of methane in methanogenic lake sediments

Hanni Vigderovich[a], Werner Eckert[b], Michal Elul[a], Maxim Rubin-Blum[c], Marcus Elvert [d], Orit Sivan[a]

[a] Department of Earth and Environmental Science, Ben-Gurion University of the Negev, Beer Sheva, Israel

[b] Israel Oceanographic & Limnological Research, The Yigal Allon Kinneret Limnological Laboratory, Israel

[c] Israel Oceanographic & Limnological Research, Haifa, Israel

[d] MARUM - Center for Marine Environmental Sciences and Faculty of Geosciences, University of Bremen, Bremen, Germany

*Corresponding author*: Hanni Vigderovich, hannil@post.bgu.ac.il

## Abstract

Anaerobic oxidation of methane (AOM) is among the main processes limiting the release of the greenhouse gas methane from natural environments. Geochemical profiles and experiments with fresh sediments from Lake Kinneret (Israel) indicate that iron-coupled AOM (Fe-AOM) sequesters 10-15% of the methane produced in the methanogenic zone (> 20-cm sediment depth). The oxidation of methane in this environment was shown to be mediated by a combination of *mcr* gene-bearing archaea and *pmoA* gene-bearing aerobic bacterial methanotrophs. Here, we used sediment slurry incubations under controlled conditions to elucidate the electron acceptors and microorganisms that are involved in the AOM process over the long term (~18 months). We monitored the process with the addition of $^{13}$C-labeled methane and two stages of incubations: (i) enrichment of the microbial population involved in AOM and (ii) slurry dilution and manipulations, including the addition of several electron acceptors (metal oxides, nitrate, nitrite and humic substances) and inhibitors (2-bromoethanesulfonate, acetylene and sodium molybdate) of methanogenesis, methanotrophy and sulfate reduction/sulfur disproportionation. Carbon isotope measurements in the dissolved inorganic carbon pool suggest the persistence of AOM, consuming 3-8% of the methane produced at a rate of $2.0\pm0.4$ nmol g$^{-1}$ dry sediment day$^{-1}$. Lipid carbon isotopes and metagenomic analyses point towards methanogens as the sole microbes performing the AOM process by reverse methanogenesis. Humic substances and iron oxides, but not sulfate, manganese, nitrate, or nitrite, are the likely electron acceptors used for this AOM. Our observations support the contrast between methane oxidation mechanisms in naturally anoxic lake sediments, with potentially co-existing aerobes and anaerobes, and long-term incubations, wherein anaerobes prevail.

Keywords: Anaerobic oxidation of methane (AOM), lake sediments, dissolved inorganic carbon, stable carbon isotopes, electron acceptors, archaea, methanogens, methanotrophs, lipids.

34

## 1. **Introduction**

Methane ($CH_4$) is an important greenhouse gas (Wuebbles and Hayhoe, 2002), which has both anthropogenic and natural sources, the latter of which account for about 50% of the emission of this gas to the atmosphere (Saunois et al., 2020). Naturally occurring methane is mainly produced biogenically via the methanogenesis process, which is performed by methanogenic archaea. Traditionally acknowledged as the terminal process anchoring carbon remineralization (Froelich et al. 1979), methanogenesis occurs primarily via the reduction of carbon dioxide by hydrogen in marine sediments and via acetate fermentation in freshwater systems (Whiticar et al. 1986).

Methanotrophy, the aerobic and anaerobic oxidation of methane (AOM) by microbes, naturally controls the release of this gas to the atmosphere (Conrad, 2009; Reeburgh, 2007; Knittel and Boetius, 2009). In marine sediments, up to 90% of the upward methane flux is consumed anaerobically by sulfate, and in established diffusive profiles, that methane consumption occurs within a distinct sulfate-methane transition zone (Valentine 2002). While sulfate-dependent AOM, catalyzed by the archaeal ANaerobic MEthanotrophs (ANMEs) 1-3, is widespread chiefly in marine sediments (Hoehler et al., 1994; Boetius et al., 2000; Orphan et al., 2001; Treude et al., 2005, 2014), methane oxidation in other environments can be coupled to other electron acceptors (e.g. Raghoebarsing et al., 2006; Ettwig et al. 2010; Sivan et al., 2011; Crowe et al. 2011; Norði and Thamdrup 2014; Valenzuela et al., 2017).

In freshwater sediments, sulfate is often depleted, and methanogenesis may be responsible for most of the organic carbon remineralization, resulting in high concentrations of methane in shallow sediments (Sinke et al., 1992). Indeed, lakes and wetlands, are responsible for 33-55% of naturally emitted methane (Rosentreter et al., 2021). A large portion of this produced methane is oxidized by aerobic (type I, type II and type X) methanotrophic bacteria via oxygen. Aerobic methanotrophy is generally observed in the sediment-water interface (Damgaard et al. 1998) and/or in the water column thermocline (Bastviken 2009). AOM, however, can also consume over 50% of the produced methane (Segarra et al. 2015).

Sulfate can be an electron acceptor of AOM in freshwater sediments, as was shown for example in Lake Cadagno (Schubert et al., 2011, Su et al., 2020). Alternative electron acceptors for AOM in natural freshwater environments and cultures include humic substances, nitrate, nitrite and metals (such as iron manganese and chromium). Natural humic substances and their synthetic analogs were shown to function as terminal electron acceptors for AOM in soils, wetlands and cultures (Valenzuela et al., 2017; 2019; Bai et al., 2019; Zhang et al., 2019; Fan et al., 2020). Nitrate-dependent AOM has been demonstrated in a consortium of archaea and denitrifying bacteria from a canal (Raghoebarsing et al., 2006), in freshwater lake sediments (Norði and Thamdrup 2014) and a sewage enrichment culture of ANME-2d (Haroon et al., 2013; Arshad et al., 2015). Nitrite is exploited to oxidize methane by the

aerobic bacteria *Methylomirabilis* (NC-10), which split the nitrite to $N_2$ and $O_2$ and then uses the
produced oxygen to oxidize the methane (Ettwig et al., 2010). ANME-2d were also suggested to be
involved in Cr(VI) coupled AOM, either alone or with a bacterial partner (Lu et al., 2016). Iron and/or
manganese coupled AOM have also been suggested in lakes (Sivan et al., 2011; Crowe et al. 2011;
Norði et al., 2013), sometimes by supporting sulfate-coupled AOM (Shubert et al., 2011; Su et al., 2020;
Mostovaya et al., 2021). Iron-coupled AOM was also shown to occur in enriched, denitrifying cultures
from sewage where it was performed by ANME-2 (Ettwig et al. 2016), and in a bioreactor with natural
sediments (Cai et al., 2018).
The mechanism and role of iron-coupled AOM in lake sediments have been studied with a variety of
tools in the sediments of Lake Kinneret. *In-situ* pore water profiles and top core experiments (Sivan et
al., 2011), diagenetic models (Adler et al., 2011) and batch incubation experiments with fresh sediment
slurries (Bar-Or et al., 2017) suggest that iron coupled-AOM (Fe-AOM) removes 10-15% of the
produced methane in the deeper part of the methanogenic zone (> 20 cm below the water-sediment
interface). Analysis of the microbial community structure suggested that both methanogenic archaea
and methanotrophic bacteria are potentially involved in methane oxidation (Bar-Or et al., 2015).
Analyses of stable isotopes in fatty acids, 16S rRNA gene amplicons and metagenomics showed that
both reverse methanogenesis by archaea and bacterial type I aerobic methanotrophy by
Methylococcales play important role in methane cycling (Bar-Or et al., 2017; Elul et al., 2021). Aerobic
methanotrophy, which has also been observed in the hypolimnion and sediments of several other lakes
that are considered anoxic (Beck et al., 2013; Oswald et al., 2016; Martinez-Cruz et al., 2017; Cabrol
et al., 2020), may be driven by the presence of oxygen at nanomolar levels (Weng et al., 2018). Pure
cultures of the ubiquitous aerobic methanotrophs Methylococcales have indeed been shown to survive
under hypoxia conditions either by oxidizing methane and with nitrate (Kits et al., 2015), by switching
to iron reduction (Zheng et al., 2020), or even by exploiting their methanobactins to generate their own
oxygen to fuel their methanotrophic activity (Dershwitz et al., 2021). The latter study also showed that
the alphaproteobacterial methanotroph *Methylocystis* sp., strain SB2, can couple methane oxidation and
iron reduction. However, whether these aerobic methanotrophic bacteria are able to oxidize methane
under strictly anoxic conditions and which electron acceptors facilitate that activity are still not known.
In the current study, we used long-term anaerobic incubations to assess the dynamics of methane-
oxidizing microbes under anoxic conditions and to quantify the respective availabilities of different
electron acceptors for AOM. To that end, we diluted fresh methanogenic sediments from Lake Kinneret
with original porewater from the same depth and amended the sediment with [13]C-labeled methane. Our
experiment design comprised of two stages, the first of which included the enrichment of the microbial
population involved in AOM, while the second involved an additional slurry dilution and several
manipulations with different electron acceptors and inhibitors. We measured methane oxidation rates
(based on [13]C-DIC enrichment), determined the characteristics of each electron acceptor (via its

turnover), and evaluated changes in microbial diversity over various incubation periods (based on metagenomics and lipid biomarkers). The results from the long-term anaerobic incubations were compared to those of batch and semi-continuous bioreactor experiments.

## 2. Methods

### 2.1 Study site

Lake Kinneret (Sea of Galilee) is a warm, monomictic, freshwater lake that is 21 km long and 13 km wide and located in northern Israel. Its maximum depth is ~42 m at its center (station A, Figure S1) while its average depth is 24 m. From March to December, the lake is thermally stratified, and from April to December, the hypolimnion is anoxic. Surface water temperatures range from 15°C in the winter (January) to 32°C in the summer (August), while the lake's bottom water temperatures remain in the range of 14-17°C throughout the year. The sediment from the deep methanogenic zone used in this study (sediment samples taken from a sediment depth of ~20 cm from the water-sediment interface at the lake's center) contains 50% carbonates, 30% clay and 7% iron (Table S1). The dissolved organic carbon (DOC) concentration of the porewater increases with depth, ranging from ~6 mg C $L^{-1}$ at the sediment-water interface to 17 mg C $L^{-1}$ at a depth of 25 cm (Adler et al., 2011). The concentrations of dissolved methane in the sediment porewater increase sharply with sediment depth, reaching a maximum of more than 2 mM at a depth of 15 cm, after which the amounts of dissolved methane gradually decreased with depth to 0.5 mM at a depth of 30 cm (Adler et al., 2011; Sivan et al., 2011; Bar-Or et al., 2015).

### 2.2 Experimental setup

2.2.1 General

In this study we compared three incubation strategies (A, B and C; Fig. 1) in Lake Kinneret methanogenic sediments (sediment depths > 20 cm), which were amended with original porewater from the same depth, $^{13}$C-labeled methane (0.05-2 ml; Table 1), different potential electron acceptors for AOM (nitrite, nitrate, iron and manganese oxides and humic substances) and activity inhibitors. We inhibited the *mcr* gene with 2-bromoethanesulfonate (BES), methanogenesis and methanotrophy with acetylene, and sulfate reduction and sulfur disproportionation with Na-Molybdate (Nollet et al., 1997; Oremland & Capone, 1988; Lovley & Klug, 1983). Below we describe the three incubation strategies (Fig. 1).

A) Long-term, two-stage slurry incubations with a 1:1 sediment to porewater ratio and high methane content for the first three months (first stage) to ensure the enrichment of the microorganisms involved in AOM. After three months, the slurry was diluted with porewater to a 1:3 ratio (second stage) and different reactants were added to the incubations, which were subsequently monitored for up to 18 months.

B) Semi-continuous bioreactor experiments in which sediments were collected up to three days before
the experiment was set up (freshly sampled sediments). The sediment to porewater ratio was 1:4 and
porewater was exchanged regularly.
C) Batch incubation experiments with freshly sampled sediments and porewater at a 1:5 ratio,
respectively, and amended with hematite. This experimental set-up was described in our previous
studies (Bar-Or et al., 2017; Elul et al., 2021).
The sediments for the slurries conducted in the current work were collected during seven day-long
sampling campaigns aboard the research vessel *Lillian* between 2017 and 2019 from the center of the
lake (Station A, Fig. S1) using a gravity corer with a 50-cm Perspex core liner. The length of the
sediment in each core was 35-45 cm. During each sampling campaign, 1-2 sediment cores were
collected for the incubations and 10 cores were collected for the porewater extraction. Sediments from
the deeper methanogenic zone (sediment depths > 20 cm) for the experiments were diluted with
porewater from the methanogenic zone of parallel cores sampled on the same day. The bottom part of
the sediment cores (below 20 cm) was transferred, as a bulk, to a dedicated 5 L plastic container
onboard. The cores and the container were brought back to the lab, where the cores were kept at 4°C,
and the porewater was extracted on the same day of sampling. In the lab, sediments were collected from
the container with 20-ml cutoff syringes and moved to 50-ml falcon tubes. The porewater was extracted
by centrifugation at 9300 g for 15 min at 4°C, syringe filtered by 0.22-$\mu$M filters into 250-ml pre-
autoclaved glass bottles, crimp-sealed with rubber stoppers, and flushed for 30 min with $N_2$. The
extracted porewater was kept under anaerobic conditions at 4°C until its use. The sediments for the
incubations were subsampled from the liners and diluted no later than three days after their collection
from the lake and treated further according to the experimental strategies described above (setup A or
B).
2.2.2 Experiment type A set-up: Long-term two-stage incubations (henceforth referred to as "two-
stage" for simplicity)
Experiment A comprised ten two-stage incubation experiments (experiment serial numbers (SN) 1-10;
Table 1) with different treatments (electron acceptors/shuttling/inhibitors). In the first stage (pre-
incubation slurry), the sediment core was sliced under continuous $N_2$ flushing and sediments from
depths > 20 cm were collected into zipper bags. The sediment was homogenized by shaking the
sediment in the bag, and between 80-100 gr was transferred into 250-ml glass bottles under continuous
$N_2$ flushing. The sediments were diluted with the extracted porewater to create a 1:1 sediment to
porewater slurry with a headspace of 70-90 ml (Fig. 1). The slurries were sealed with rubber stoppers
and crimped caps and were flushed with $N_2$ (99.999%, MAXIMA, Israel) for 30 min. Methane (99.99%,
MAXIMA, Israel) was injected using a gas-tight syringe for a final content of 20% in the headspace,
where 10% of the injected methane was $^{13}$C-labeled methane (99%, Sigma-Aldrich). When significant
AOM activity was observed based on the increase of $\delta^{13}C_{DIC}$ after approximately three months (Fig.
S2), some of the incubations were further diluted during the second stage of the experiments. The
remainder of the incubations continued to be run with porewater exchange while the $\delta^{13}C_{DIC}$ values
were monitored every three months.
All the experiments were set up similarly (see dates and detailed protocols in the supplementary
information): the pre-incubation bottle was opened and subsamples (~18 g each) were transferred with
a syringe and a Tygon® tube under a laminar hood and continuous flushing of $N_2$ gas into 60-ml glass
bottles. The subsamples were then diluted with fresh anoxic porewater from the methanogenic zone (as
described above) to achieve a 1:3 sediment to porewater ratio (Fig. 1) while leaving 24 ml of headspace
in each bottle. The bottles were crimp-sealed, flushed with $N_2$ gas for 5 min, shaken vigorously and
flushed again (3 times). Then $^{13}C$-labeled methane was added to all of the bottles as described in Table
1. The "killed" control slurries in each experiment were autoclaved twice and cooled, only after which
they were amended with the appropriate treatments and $^{13}C$-labeled methane.
To the diluted (1:3) batch slurries electron acceptors were added either as a powder (hematite –
experiment no. 1, magnetite – experiment no. 2, clay and humic substances – experiment no. 7, $MnO_2$
– experiment no. 3) or in dissolved form in double-distilled water (DDW) ($KNO_3$ – experiment no. 4,
$NaNO_2$ – experiment no. 5). In addition, the potential involvement of sulfur cycling in the transfer of
electrons was tested in experiment no. 2 via its inhibition with Na-molybdate (Lovley and Klug, 1983).
The synthetic analog for humic substances, i.e., 9,10-anthraquinone-2,6-disulfonate (AQDS), was
dissolved in DDW (detailed in the supplementary information) and added to the bottles of experiment
no. 6 until a final concentration of 5 mM was achieved in each bottle. Amorphous iron ($Fe(OH)_3$) was
prepared in the lab by dissolving $FeCl_3$ in DDW that was then titrated with NaOH 1.5 N up to pH 7 and
injected into the bottles of experiment no. 2. The final concentration of each addition is detailed in Table
1. The $^{13}C$-labeled methane was injected into all of the experimental bottles at the beginning of each
experiment (unless described otherwise) by using a gas-tight syringe from a stock bottle filled with $^{13}C$-
labeled methane gas (which was replaced with saturated NaCl solution). Three different inhibitors were
added to three different experiments: Molybdate was added to experiment No. 1 (to one bottle of
methane-only treatment, magnetite treatment and amorphous iron treatment) to detect the feasibility of
an active sulfur cycle; BES was added to experiment No. 8 at the start of the experiment; and acetylene
was added to experiment No. 9, wherein it was injected during the experiment into two bottles at
different timepoints after $^{13}C$ enrichment was observed in the DIC (Table 1).
All live treatments were set up in duplicate or triplicate, depending on the amount of the pre-incubated
slurry aimed for each experiment, and the results are presented as the average with an error bar. In two
experiments, only one "killed" control bottle was set up, and the remainder of the slurry was prioritized
for other treatments because the killed controls repeatedly showed no activity in several previous
experiments. The humic substrate experiment used a natural (humic) substance that was extracted from
a lake near Fairbanks, Alaska, where iron reduction was observed in the methanogenic zone. One
experiment was set up without any additional electron acceptor to assess the rate of methanogenesis in
the two-stage slurries. Porewater was sampled anaerobically for $\delta^{13}C_{DIC}$ and dissolved Fe(II)
measurements in duplicate (2 ml), and methane was measured from the headspace. Variations in the
$\delta^{13}C_{DIC}$ values between the experiments resulted from different amounts of $^{13}$C-labeled methane injected
at the start of each experiment (geochemical measurements detailed in the analytical methods section
below).
2.2.3 Experiment type B setup: Semi-continuous bioreactor
Semi-continuous bioreactors were used to monitor the redox state regularly at close-to-natural *in-situ*
conditions for 15 months in freshly collected sediments. Two 0.5-L semi-continuous bioreactors (Fig.
1) (LENZ, Weinheim, Germany) were set up with freshly sampled sediments from the methanogenic
zone (25 - 40 cm) and extracted porewater from the same depth from Station A on Lake Kinneret
immediately after their collection. Both reactors were filled, headspace-free, with a slurry at a 1:4
sediment to porewater ratio. One bioreactor was amended with 10 mM hematite while the second, which
was a control, was not amended. To dissolve $^{13}$C-labeled methane in the porewater, 15 ml of porewater
were replaced with 15 ml of methane gas (13 ml of $^{12}CH_4$ and 2 ml of $^{13}CH_4$) to produce a methane-
only headspace for 24 h, during which time the reactors were shaken repeatedly. After 24 h, the gas was
replaced with anoxic porewater, thus eliminating the headspace, which resulted in lower methane
concentrations (0.2 mM) than in either the two-stage incubations or the fresh batch experiment (~2 m).
The redox potential was monitored continuously using a platinum/glass electrode (Metrohm, Herisau,
Switzerland) to verify anoxic conditions and to determine the redox state throughout the incubation
period. The bioreactors were subsampled weekly to bi-weekly, and the sample volume (5-10 ml) was
replaced immediately by preconditioned anoxic (flushed with $N_2$ gas for 15 min) porewater from the
methanogenic zone. As outlined below, samples were analyzed for dissolved Fe(II), methane and
$\delta^{13}C_{DIC}$. Additional subsamples for metagenome and lipid analyses were taken at the beginning of the
experiment and on days 151 and 382, respectively.
2.2.4 Experiment type C setup: Fresh batch experiment
Sediments for this experiment were collected in August 2013 at Station A using a protocol similar to
that used to collect the sediments for the pre-incubations. Sediments from depths greater than 26 cm
were diluted under anaerobic conditions with porewater from the same depth to obtain a ratio of
sediment to porewater of 1:5. The resulting slurry was then divided between 60-ml glass bottles (40 ml
slurry in each bottle). The sampling and experimental setup are described in detail in our earlier study
(Bar-Or et al., 2017). Here we present our results of the $\delta^{13}C_{DIC}$, metagenome and lipid analyses of two
treatments: natural (with only $^{13}$C-labeled methane) and hematite. The experiment ran for 15 months.

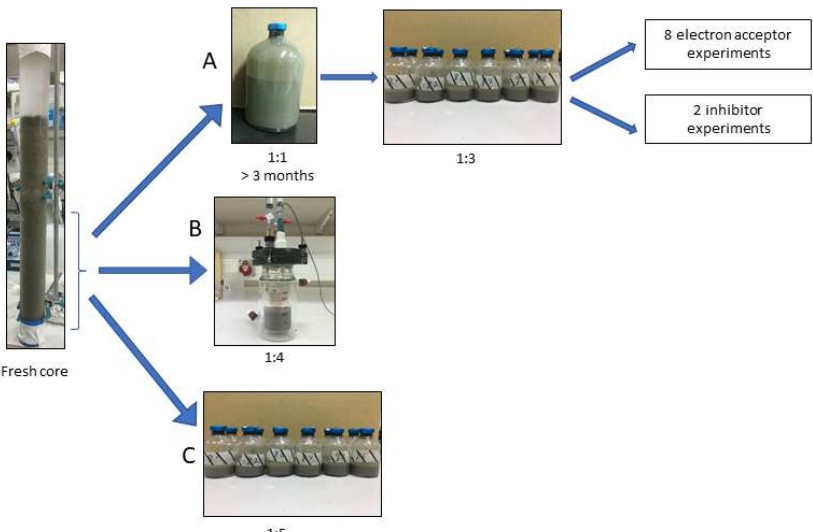


Figure 1: Flow diagram of the experimental design. Three types of experiments were set up to investigate the
methanogenic zone sediments (deeper than 20 cm): **A)** Two-stage slurry experiments, with 1:1 ratio of sediment
to porewater incubations and then with diluted pre-incubated slurries and porewater (1:3 ratio of sediment to
porewater). **B)** Semi-continuous bioreactor experiment with freshly collected sediment. **C)** Fresh batch
experiment – slurry experiment with freshly collected sediments (Bar-Or et al., 2017).

Table 1: Details of the three types of experiments: two-stage, semi-aerobic bioreactor and fresh batch experiments.

| Experiment serial number (SN) | Experiment | Treatment | # of bottles | CH4 [ml] | 13CH4 [ml] | Fe2O3 [mM] | Fe3O4 [mM] | Fe(OH)3 [mM] | MnO2 [mM] | NO2- [mM] | NO3- [mM] | AQDS [mM] | Humic substances [mM] | PCA [mM] | Fe-bearing nontronite (clay) [gr] | Na2-molybdate [mM] | BES [mM] | Acetylene [µL] | Temp [c°] | Duration [day] | Comments |
|---|---|---|---|---|---|---|---|---|---|---|---|---|---|---|---|---|---|---|---|---|---|
| 1 | Hematite | 13CH4 | 2 | | 1 | | | | | | | | | | | | | | 20 | 201 | |
| | | 13CH4+hematite | 2 | | 1 | 10 | | | | | | | | | | | | | 20 | 201 | The methane that was added at the beginning of the experiment was not labelled, so 13C-labeled methane was added after 105 days. Na2–molybdate was added to one of the bottles on day 365 |
| 2 | Magnetite | 13CH4 | 2 | | 1 | | | | | | | | | | | | | | 16 | 447 | Na2–molybdate was added to one of the bottles on day 365 |
| | | 13CH4+magnetite | 2 | | 1 | | 10 | | | | | | | | 1 | | | | 16 | | |
| | | 13CH4+Fe(OH)b | 2 | | 1 | | | 10 | | | | | | | 1 | | | | 16 | | |
| | | Killed+13CH4+magnetite | 1 | | 1 | | 10 | | | | | | | | 1 | | | | 16 | | |
| 3 | MnO2 | 13CH4 | 2 | | 1.2 | | | | | | | | | | | | | | 20 | 201 | 200 µL 13CH4 was added on day 1, then another 1 mL was added on day 24. |
| | | 13CH4+MnO2 | 2 | | 1.2 | | | | 10 | | | | | | | | | | 20 | | 200 µL 13CH4 was added on day 1, then another 1 mL was added on day 24. |
| 4 | Nitrate | 13CH4+NO3 (high conc.) | 2 | 1 | 0.5 | 12 | | | | | 1 | | | | | | | | 20 | 306 | |
| | | 13CH4+hematite | 2 | 1 | 0.5 | 12 | | | | | | | | | | | | 20 | | |
| | | 13CH4+NO3 (high conc.)+Hematite | 2 | 1 | 0.5 | 12 | | | | | 1 | | | | | | | 20 | | |
| | | 13CH4+NO3 (low conc.)+hematite | 2 | 1 | 0.5 | 12 | | | | | 0.2 | | | | | | | 20 | | |
| | | Killed+13CH4+NO3 (high conc.)+hematite | 1 | 1 | 0.5 | 12 | | | | | 1 | | | | | | | 20 | | |
| 5 | Nitrite | 13CH4 | 3 | 1 | 0.5 | | | | | | | | | | | | | 20 | 493 | |
| | | 13CH4+NO2 (high conc.)+Hematite | 2 | 1 | 0.5 | 10 | | | | 0.5 | | | | | | | | 20 | | |
| | | 13CH4+NO2 (low conc.)+hematite | 2 | 1 | 0.5 | 10 | | | | 0.1 | | | | | | | | 20 | | |
| | | Killed+13CH4+NO2 (high conc.)+hematite | 2 | 1 | 0.5 | 10 | | | | 0.5 | | | | | | | | 20 | | |
| 6 | AQDS | 13CH4 | 3 | 1 | 1 | | | | | | | | | | | | | 20 | 264 | |
| | | 13CH4+AQDS | 2 | 1 | 1 | | | | | | | 5 | | | | | | 20 | | |
| | | 13CH4+AQDS+hematite | 2 | 1 | 1 | 10 | | | | | | 5 | | | | | | 20 | | |
| | | Killed+13CH4+AQDS | 2 | 1 | 1 | | | | | | | | | | | | | 20 | | |
| 7 | Natural humic acids and clay | 13CH4 | 2 | 1 | 1 | | | | | | | | | | | | | 20 | 169 | The head space of the experiment bottles was flushed with N2 on day 51 and 13CH4 was added. This was done in order to match the the clay bottles. |
| | | 13CH4+hematite | 2 | 1 | 1 | 10 | | | | | | | | | | | | 20 | | |
| | | 13CH4+humic acid | 2 | 1 | 1 | | | | | | | | 0.5 | | | | | 20 | | |
| | | 13CH4+clay | 2 | | 1 | | | | | | | | | | 1 | | | 20 | | Clay was added on day 43, and the bottles were flushed again with N2, 13CH4 was added again on day 51. |
| | | Killed+13CH4+hematite | 2 | | 1 | 10 | | | | | | | | | | | | 20 | | |
| 8 | Bromoethanesulfonate (BES) | 13CH4+hematite | 2 | 9 | 1 | 10 | | | | | | | | | | | | 20 | 493 | |
| | | 13CH4+hematite+BES | 2 | 9 | 1 | 10 | | | | | | | | | | 20 | | 20 | | |
| 9 | Acetylene | 13CH4+hematite | 4 | 1 | 0.5 | 10 | | | | | | | | | | | | 20 | 321 | Acetylene was injected to each bottle at different time point during the experiment. |
| | | 13CH4+hematite+acetylene | 2 | 1 | 0.5 | 10 | | | | | | | | | | | 120 | 20 | | |
| | | Killed+13CH4+hematite | 2 | 1 | 0.5 | 10 | | | | | | | | | | | 120 | 20 | | |
| 10 | No electron acceptor | No additions | 3 | | | | | | | | | | | | | | | 20 | 147 | |
| | | 13CH4 | 3 | | 1 | | | | | | | | | | | | | 20 | | |
| | Semi-bioreactor | 13CH4 | | 13 | 2 | | | | | | | | | | | | | 16 | 345 | |
| | | 13CH4+hematite | | 13 | 2 | 10 | | | | | | | | | | | | 16 | 677 | |
| | Freshly collected sediment exp. | 13CH4 | | 1 | 0.05 | | | | | | | | | | | | | 20 | 467 | |
| | | 13CH4+hematite | | | 0.05 | 20 | | | | | | | | | | | | 20 | | |

## 2.3 Analytical methods

2.3.1 Geochemical measurements

Measurements of $\delta^{13}C_{DIC}$ were performed on a DeltaV Advantage Thermo Scientific isotope-ratio mass-spectrometer (IRMS). Results are reported referent to the Vienna Pee Dee Belemnite (VPDB) standard. For these measurements, about 0.3 ml of filtered (0.22 μm) porewater was injected into a 12-ml glass vial with a He atmosphere and 10 μl of $H_3PO_4$ 85% to acidify all the DIC species to $CO_2$ (g). The headspace autosampler (CTC Analytics; Type PC PAL) sampled the gas from the vials and measured the $\delta^{13}C_{DIC}$ of the sample on the GasBench interface with a precision of ±0.1 ‰. DIC was measured on the IRMS using the peak height and a precision of 0.05 mM. Dissolved Fe(II) concentrations were determined using the ferrozine method (Stookey, 1970) by HANON i2 visible spectrophotometer at a 562-nm wavelength with a detection limit of 1 μmol $L^{-1}$. A 100-μL headspace sample was taken for methane measurements with a gas-tight syringe and was analyzed by gas chromatograph (Focus GC, Thermo) equipped with a flame ionization detector (FID) and a packed column (Shincarbon ST) with a helium carrier gas (UHP) and a detection limit of 1 nmol methane. Bottles to which acetylene was added were also measured by the GC with the same column and carrier gas for ethylene to determine the acetylene turnover with the N cycle.

2.3.2 Lipid analysis

A sub-set of samples (Table 3) was investigated for the assimilation of $^{13}C$-labeled methane into polar lipid-derived fatty acids (PLFAs) and intact ether lipid-derived hydrocarbons. A total lipid extract (TLE) was obtained from 0.4 to 1.6 g of the freeze-dried sediment or incubated sediment slurry using a modified Bligh and Dyer protocol (Sturt et al., 2004). Before extraction, 1 μg of 1,2-diheneicosanoyl-*sn*-glycero-3-phosphocholine and 2-methyloctadecanoic acid were added as internal standards. PLFAs in the TLE were converted to fatty acid methyl esters (FAMEs) using saponification with KOH/MeOH and derivatization with $BF_3$/MeOH (Elvert et al., 2003). Intact archaeal ether lipids in the TLE were separated from the apolar archaeal lipid compounds using preparative liquid chromatography (Meador et al., 2014) followed by ether cleavage with $BBr_3$ in dichloromethane forming hydrocarbons (Lin et al., 2010). Both FAMEs and ether-cleaved hydrocarbons were analyzed by GC-mass spectrometry (GC-MS; Thermo Finnigan Trace GC coupled to a Trace MS) for identification and by GC-IRMS (Thermo Scientific Trace GC coupled via a GC Isolink interface to a Delta V Plus) to determine $\delta^{13}C$ values by using the column and temperature program settings described by Aepfler et al. (2019). The $\delta^{13}C$ values are reported with an analytical precision better than 1‰ as determined by long-term measurements of an *n*-alkane standard with known isotopic composition of each compound. Reported fatty acid isotope data are corrected for the introduction of the methyl group during derivatization by mass balance calculation similar to equation 1 (see below) using the measured $\delta^{13}C$ value of each FAME and the known isotopic composition of methanol as input parameters.

2.3.3 Metagenomic analysis
For the metagenomic analyses, total genomic DNA was extracted from the semi-aerobic bioreactor with
hematite addition (duplicate samples), pre-incubation slurries ($^{13}CH_4$-only control, $^{13}CH_4$ + hematite)
and their respective initial slurries (t0) by using the DNeasy PowerLyzer PowerSoil Kit (QIAGEN).
Genomic DNA was eluted using 50 µl of elution buffer and stored at −20 °C. Metagenomics libraries
were prepared at the sequencing core facility at the University of Illinois at Chicago using the Nextera
XT DNA library preparation kit (Illumina, USA). Between 19 and 40 million 2 × 150 bp paired-end
reads per library were sequenced using Illumina NextSeq500. Metagenomes were co-assembled from
the concatenated reads of all of the metagenomic libraries with Spades V3.12 (Bankevich et al., 2012;
Nurk et al., 2013) after decontamination, quality filtering (QV= 10) and adapter-trimming with the
BBDuk tool from the BBMap suite (Bushnell B, http://sourceforge.net/projects/bbmap/). Downstream
analyses, including reading coverage estimates, automatic binning with maxbin (Wu et al., 2014) and
metabat2 (Kang et al., 2019) bin refining with the DAS tool (Sieber et al., 2018), were performed within
the SqueezeMeta framework (Tamames and Puente-Sánchez, 2019). GTDB-Tk was used to classify the
metagenome-assembled genomes (MAGs) based on Genome Taxonomy Database release 95 (Parks et
al., 2021). The principal component analysis biplot was constructed with Past V4.03 (Hammer et al.,
305    2001).

2.3.4 Rate calculations
Methanogenesis rates were calculated from temporal changes in methane concentration in a
representative pre-incubated slurry experiment (Fig. 2). The amount of methane oxidized was calculated
by a simple mass balance calculation according to equations 1 and 2:
$x \times F^{13}CH_4 + (1 - x) \times FDI^{13}C_i = FDI^{13}C_f$ (1)
$[CH_4]_{ox} = x \times [DIC]_f$ (2)
The final DIC pool comprises two end members, the initial DIC pool and the oxidized $^{13}C$-$CH_4$. The
term $x$ denotes the fraction of oxidized $^{13}C$-$CH_4$, while 1-$x$ denotes the fraction of the initial DIC pool
out of the final DIC pool. $F^{13}CH_4$ is the fraction of $^{13}C$ out of the total $CH_4$ at t0 (i-initial), $FDI^{13}C_i$ is
the fraction of $^{13}C$ out of the total DIC at t0, and $FDI^{13}C_f$ is the fraction of $^{13}C$ out of the total DIC at
t-final. $[CH_4]_{ox}$ is the amount (concentration in pore water) of the methane oxidized throughout the full
incubation period, and $[DIC]_f$ is the DIC concentration at t-final. It was assumed that the isotopic
composition of the labeled $CH_4$ did not change significantly throughout the incubation period.
**3. Results**
In ten sets of slurry incubation experiments, we followed the progress of the methane oxidation process
in Lake Kinneret methanogenic sediments in type A two-stage long-term incubations. This is by
monitoring the changes in $\delta^{13}C_{DIC}$ values and by running metagenomic and specific isotope lipid
analyses. We also followed methane oxidation in a semi-continuous bioreactor system (type B) with
freshly collected sediments with or without the addition of hematite (Fig. 3). The results were compared
to those of fresh batch slurry incubations (type C) from the same methanogenic zone, presented by Bar-
Or et al. (2017) and Elul et al. (2021).
**3.1 Geochemical trends in the long-term two-stage experiments**
In the second stage (1:3 ratio of sediment to porewater) long-term batch slurry experiments (type A)
from the methanogenic zone, methanogenesis occurred with net methanogenesis rates of ~ 25 nmol g
dry weight (DW)$^{-1}$ d$^{-1}$ (Fig. 2, Table S2), which are similar to those of fresh incubation experiments
(Bar-Or et al., 2017). At the same time there was a conversion of $^{13}C$-methane to $^{13}C$-DIC in all the non-
killed slurries amended with $^{13}C$-methane, indicating AOM (Figs. 3 and 4). The $\delta^{13}C_{DIC}$ values of the
"methane-only" control slurries reached as high values as 743‰. The average AOM rate in the
methane-only controls was 2.0±0.4 nmol g DW$^{-1}$ d$^{-1}$ (Table 2). AOM was observed in these geochemical
experiments also with the addition of electron acceptors, and the potential of several electron acceptors
to perform and stimulate the AOM process is detailed below.

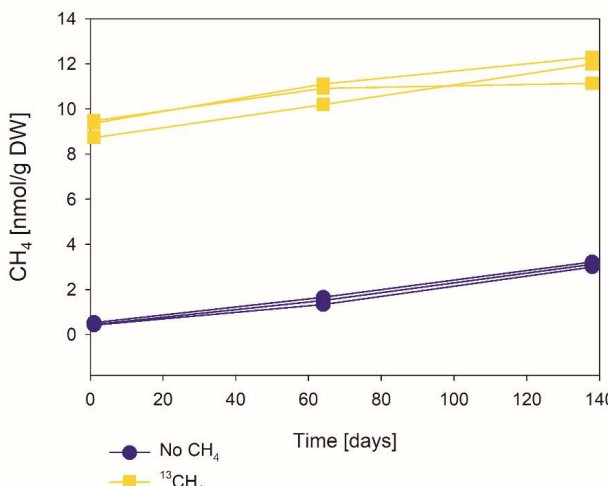


Figure 2: The change of methane concentrations with the time of a representative incubated second stage long-
term slurry experiment, showing apparent net methanogenesis with the average rate of 25 nmol g DW$^{-1}$ d$^{-1}$.
3.1.1 Metals as electron acceptors
Iron and manganese oxides were added as potential electron acceptors to the second-stage long-term
slurries. The addition of hematite to three different experiments increased the $\delta^{13}C_{DIC}$ values over time
to 694‰, similar to the behavior of the methane-only controls, and in a different pattern than the fresh
experiments (Fig. 3). The average AOM rate in those two-stage treatments was 1.0±0.3 nmol g DW$^{-1}$
d$^{-1}$ (Table 3). Magnetite amendments resulted in a minor increase of $\delta^{13}C_{DIC}$ values compared to the
methane-only controls (200‰ and 265‰, respectively, Fig. 4A) with an AOM rate of 1.8 nmol g DW$^{-}$
$^{1}$ d$^{-1}$. Amorphous iron amendments resulted in only a 22‰ increase in $\delta^{13}C_{DIC}$ and a lower AOM rate
(0.1 nmol g DW$^{-1}$ d$^{-1}$, Fig. 4A and Table 2). The addition of iron-bearing clay nontronite did not cause
any increase in the $\delta^{13}C_{DIC}$ values (Fig. 4B), but the concentration of dissolved Fe(II) increased
compared to the natural methane-only control (Fig. 5). Based on $\delta^{13}C_{DIC}$ estimates, no AOM was
detected 200 days after the addition of $MnO_2$ whereas the $\delta^{13}C_{DIC}$ values of the methane-only controls
increased to over 500‰ (Fig. 4F).
3.1.2 Sulfate as an electron acceptor
The involvement of sulfate in the AOM in the incubations was tested, even in the absence of detectable
sulfate in the methanogenic sediments. This is as sulfate could theoretically still be a short living
intermediate for the AOM process in an active cryptic sulfur cycle (Holmkvist et al., 2011). It was
quantified directly by adding Na-molybdate to the methane-only controls and the magnetite amended
treatments in the second stage long-term incubations (Fig. 4A). The addition of Na-molybdate did not
affect the increasing trend of $\delta^{13}C_{DIC}$ with time, and therefore, the AOM rates remained unchanged,
similar to the observation in the fresh batch incubations (Bar-Or et al., 2017).
3.1.3 Nitrate and nitrite as electron acceptors
Nitrate and nitrite involvement in the AOM was tested for the feasibility of an active cryptic nitrogen
cycle, even in the absence of detectable amounts of nitrate and nitrite in the sediments (Nüsslein et al.,
2001; Sivan et al., 2011). Nitrate was added at two different concentrations (0.2 and 1 mM, Fig. 4C) to
the second stage long-term slurries amended with hematite, as these concentrations were shown
previously to promote AOM in other settings (Ettwig et al., 2010). The addition of hematite alone
increased the $\delta^{13}C_{DIC}$ values by ~200‰ during the 306 days of the experiment. The $\delta^{13}C_{DIC}$ in the bottles
with the addition of 1 mM nitrate, with and without hematite (Fig. 4C; the data points of the two
treatments are on top of each other), decreased from 43‰ at the beginning of the experiment to 35‰
after 306 days. The $\delta^{13}C_{DIC}$ in the bottles with the addition of 0.2 mM nitrate and hematite increased by
27‰ at the end of the experiment. Following the addition of 0.5 mM of nitrite, we observed no increase
in $\delta^{13}C_{DIC}$ values during the first 222 days (Fig. 4D), after which they increased from 34‰ to 54‰ by
the end of the experiment. The AOM rate of the high nitrite concentration treatment was 0.2 nmol g
DW$^{-1}$ d$^{-1}$ (Table 2). Following the addition of 0.1 mM nitrite, $\delta^{13}C_{DIC}$ increased only after 130 days to
158‰ on day 493. The AOM rate of the low nitrite concentration treatment was 0.5 nmol g DW$^{-1}$ d$^{-1}$.
In the methane-only controls, the $\delta^{13}C_{DIC}$ value reached a maximum of 330‰.
3.1.4 Organic compounds as electron acceptors
Two of the second stage long-term incubation experiments were amended with synthetic and natural
organic electron acceptors to test the potential of organic electron acceptors. The addition of AQDS to
slurries with and without hematite caused a decrease in $\delta^{13}C_{DIC}$ values over the entire duration of the
experiment (Fig. 4E). Dissolved Fe(II) increased by 50 µM in these treatments, while in those without
AQDS, it exhibited an increase of 20 µM (Fig. S3). We further tested the effect of naturally occurring
humic substances by using those isolated from a different natural lake. The results show that the $\delta^{13}C_{DIC}$
values did not change at the beginning of the experiments (Fig. 4B), while a steep increase of ~90 µM
in their Fe(II) concentration was observed (Fig. 5). After 20 days, the $\delta^{13}C_{DIC}$ values of these slurries
started to increase dramatically from 84‰ to 150‰ with an AOM rate of 1.2 nmol g DW$^{-1}$ d$^{-1}$ (Fig. 4B,
Table 2). Dissolved Fe(II) concentrations mirrored the trend of $\delta^{13}C_{DIC}$ with a steep increase during the
first 20 days followed by a decrease of 37 µM (Fig. 5).
3.1.5 Metabolic pathways
To elucidate which metabolic processes drive AOM, we analyzed $\delta^{13}C_{DIC}$ following the addition of
inhibitors to the second stage long-term slurries: i) BES, a specific inhibitor for methanogenesis (Nollet
et al., 1997) and ii) acetylene, a non-specific inhibitor for methanogenesis and methanotrophy
(Orembland and Capone, 1988). In both cases and similar to the killed control, labeled $^{13}$C-DIC
production was completely inhibited following the addition (Fig. 6). Though acetylene can also inhibit
nitrogen cycling in some cases, it has been shown to result in the production of ethylene (Oremland and
Capone, 1988). In our case, however, no ethylene was detected, supporting the conclusion that only the
methanogenesis activity was inhibited.

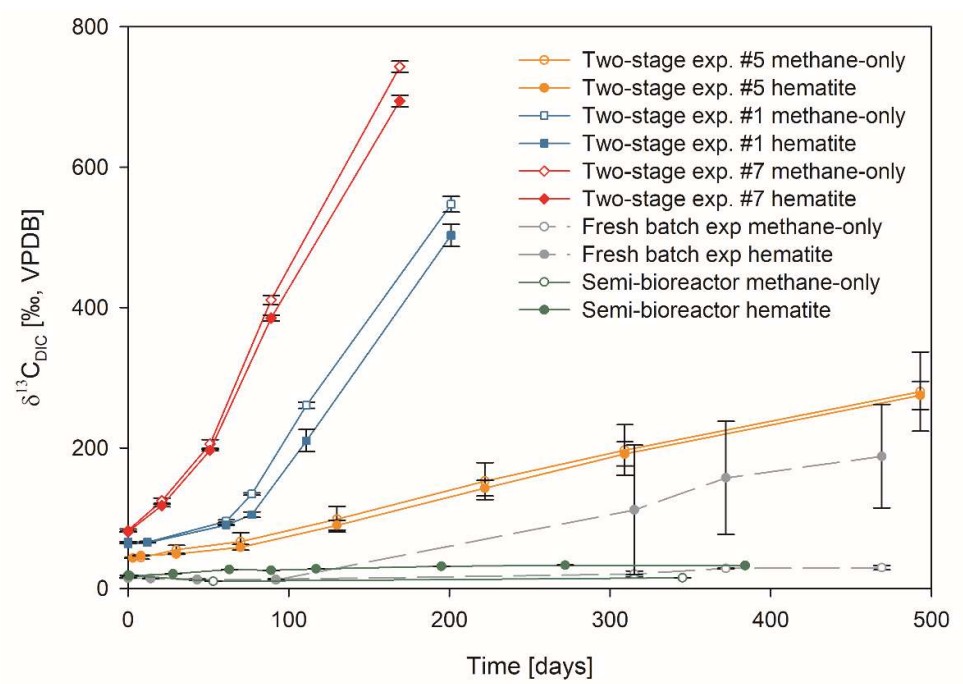


Figure 3: Comparison of $\delta^{13}C_{DIC}$ values among the three types of experiments with $^{13}C$ -labeled methane addition:
A) three two-stage slurry experiments (at the second stage of 1:3 ratio of sediment to porewater); B) the semi-
continuous bioreactor experiment; and C) slurry batch experiment with freshly collected sediments (Bar-Or et al.,
2017). In each experiment, two treatments are shown, with hematite (filled symbol) and without hematite (empty
symbols). The error bars represent the average deviation of the mean of duplicate/triplicate bottles.

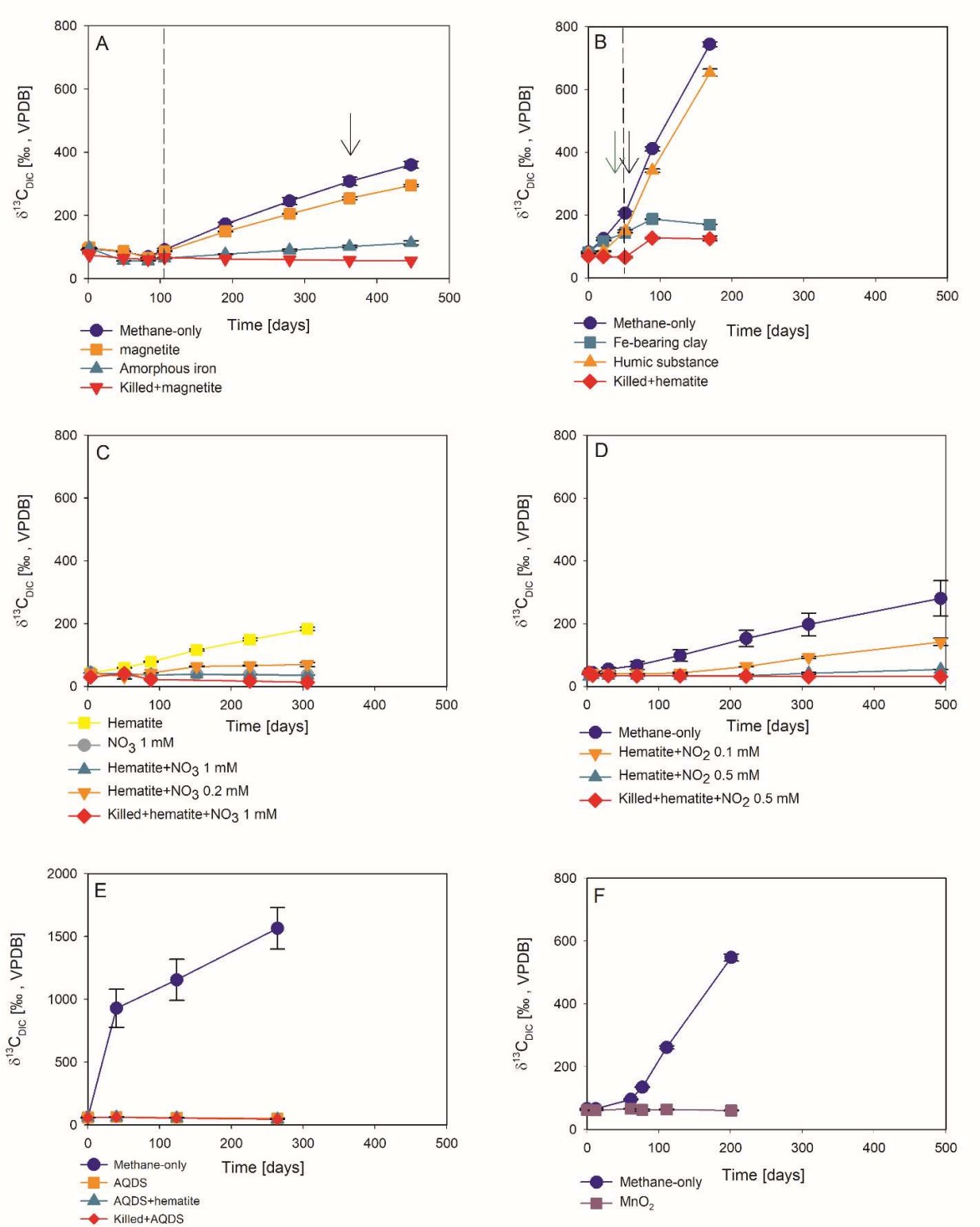


Figure 4: Potentials of different electron acceptors for AOM in Lake Kinneret in the two-stages long-term slurry experiments (at the second stage of 1:3 ratio of sediment to porewater) with of $^{13}C$ -labeled methane and the following treatments: (A) with and without the addition of magnetite and amorphous iron (Fe(OH)$_3$). The dashed line represents the specific time of $^{13}C$ -labeled methane addition. The black arrow represents the addition of Na-molybdate as an inhibitor for sulfate reduction. (B) with clay and natural humic substance. The green arrow represents the time clay was added to the relevant bottles, the dashed line represents the time the headspace of each bottle was flushed again with N$_2$, and the black arrow represents the second injection of 1 mL of $^{13}C$-labeled methane. (C) with the addition of hematite and two different concentrations of nitrate. (D) with the addition of hematite and two different concentrations of nitrite. (E) with the addition of AQDS. (F) with and without the addition of $^{13}C$-labeled methane to all the bottles (see Table 1 for specific experimental details). Error bars represent the average deviations of the data points from their means of duplicate/triplicate bottles.

Table 2: AOM rates and AOM role in experiment type A second stage slurries amended with $^{13}C$-labeled methane and different electron acceptors (assuming methanogenesis rate of 24.8 nmol g DW$^{-1}$ d$^{-1}$).

| Experiment serial number (SN) | Treatment | AOM rate [nmol/g DW X d] | AOM/methanogenesis [%] |
|---|---|---|---|
| 10 | methane only | 1.1 | 4.4 |
| 1 | methane only | 1.6 | 6.4 |
| | methane+hematite | 0.5 | 2.1 |
| 2 | methane only | 2.4 | 8.2 |
| | methane+magnetite | 1.8 | 6.3 |
| | methane+amorphous iron | 0.1 | 0.5 |
| 7 | methane only | 1.4 | 6.4 |
| | methane+hematite | 1.3 | 6.0 |
| | methane+humics | 1.2 | 5.4 |
| 5 | methane only | 1.0 | 4.6 |
| | methane+hematite | 1.0 | 4.6 |
| | methane+hematite+nitrite 0.5 mM | 0.2 | 0.8 |
| | methane+hematite+nitrite 0.1 mM | 0.5 | 2.1 |

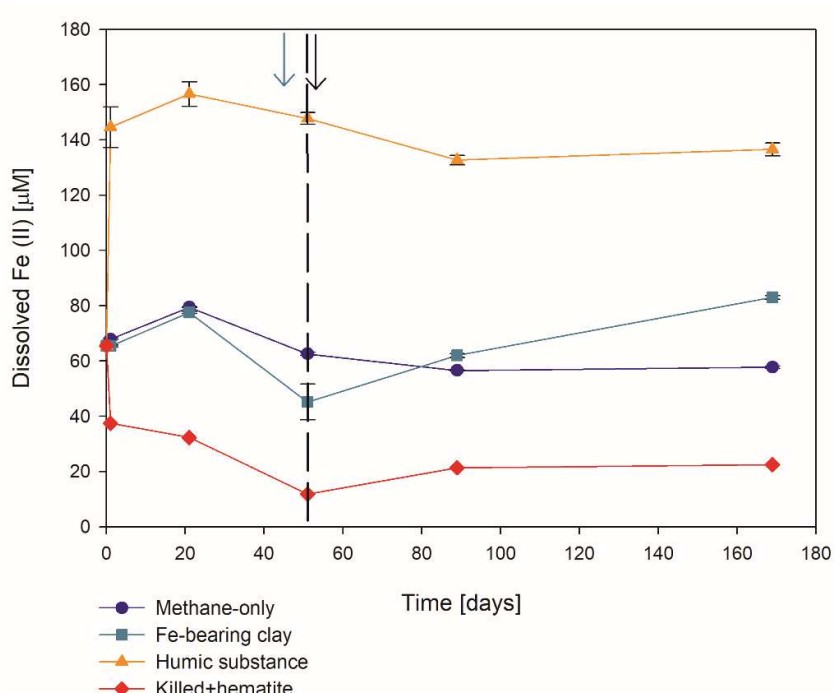


Figure 5: Change in dissolved Fe(II) in the second stage of experiment No. 7 containing clay and natural humic
acid. The green arrow represents the time at which clay was added to the specific bottles and those bottles were
flushed with $N_2$, the dashed line represents the time at which the rest of the bottles were flushed, and the black
arrow represents the time at which $^{13}C$-labeled methane was added again. Error bars represent the average of the
absolute deviations of the data points from their means.

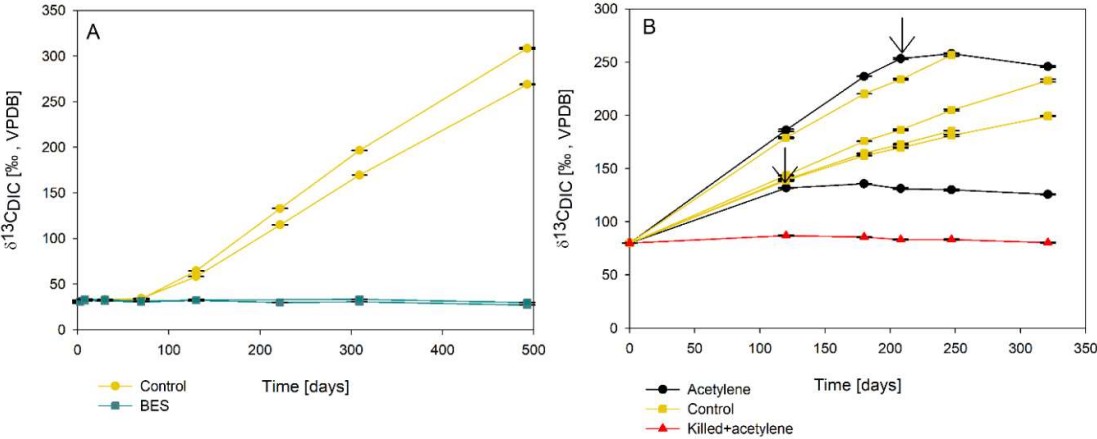


Figure 6: Change in $\delta^{13}C_{DIC}$ values over time in the second stage long-term sediment slurry incubations amended
with hematite and $^{13}C$-labeled methane. (A) with/without BES and (B) with/without acetylene. Black arrows
represent the time at which acetylene was injected into the experiment bottle. The error bars are smaller than the
symbols.

**3.2 Microbial dynamics**

Analyses of taxonomy and coverage of metagenome-assembled genomes suggest that in the pre-incubated two-stage slurries, Bathyarchaeia are the dominant archaea, together with putative methanogens such as Methanofastidiales (Thermococci), Methanoregulaceae (Methanomicrobia) and Methanotrichales (Methanosarcina) (Supplementary coverage table). Bona-fide ANME (ANME-1) were detected with substantial coverage of approximately 1 (the 27[th] most abundant from among the 195 MAGs detected) in all of the treatments. Among the bacteria, the sulfate reducers Desulfobacterota and Thermodesulfovibrionales (Nitrospirota) were prominent together with the GIF9 Dehalococcoida lineage, which is known to metabolize chlorinated compounds in lake sediments (Biderre-Petit et al., 2016). Some Methylomirabilales (NC10) were found (average coverage of 0.32±0.06), and no Methanoperedens were detected. Methylococcales methanotrophs were found in the natural sediments and the fresh batch and bioreactor incubations (average of 0.34±0.02), in contrast to their average coverage of 0.09±0.04 in the long-term incubations. Methylococcales comprised the *Methyloterricola*, *Methylomonas* and *Methylobacter* genera (Supplementary coverage table). The methylotrophic partners of aerobic methanotrophs, *Methylotenera*, were found in fresh batch and bioreactor incubations, where *Methylomonas* was found, findings that are in line with those of previous studies that showed their association (Beck et al., 2013). Principal component analysis shows the grouping of long-term, pre-incubated slurries, semi-aerobic bioreactor incubations, and fresh batch experiments (Fig. 7), emphasizing the microbial dynamics over time.

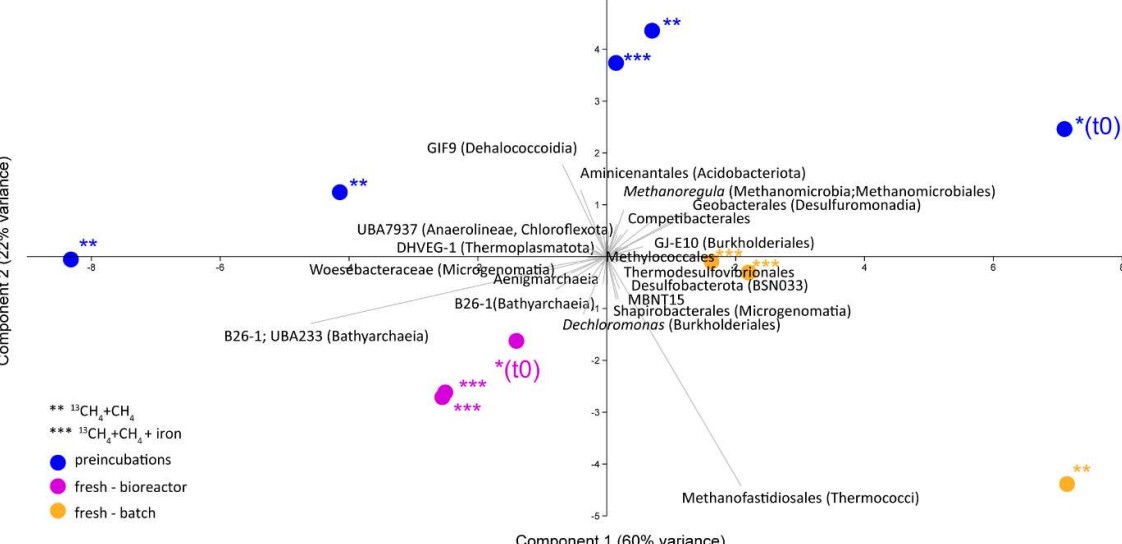

Figure 7: Principal component analysis comparison of three types of samples: long-term pre-incubated slurries (blue – experiment A), semi-continuous bioreactor (pink – experiment B) and fresh batch experiments (orange – experiment C). One asterisk represents t0, two asterisks denote methane-only treatments, three asterisks represent hematite treatment.

454 **3.3 Lipid analysis**

The $\delta^{13}C$ values of the archaeol-derived isoprenoid phytane were between −5 and −17‰ in the long-
term pre-incubated samples and thus showed $^{13}C$-enrichment of 15 to 27‰ relative to the original
sediment. This is indicative of methane-derived carbon assimilation by archaea (Table 3). Acyclic
biphytane, derived mainly from caldarchaeol, exhibited a less pronounced $^{13}C$-enrichment of 5-10‰.
For bacterial-derived fatty acids, $\delta^{13}C$-values similarly shifted by up to 10‰ relative to the original
sediment. Nonetheless, one would have expected much higher values if aerobic methanotrophs were
active, as was previously indicated by strong $^{13}C$-enrichments of up to 1,650‰ in $C_{16:1\omega5c}$ observed in
freshly incubated batch samples (Bar-Or et al., 2017).
Table 3: The $\delta^{13}C$ values (in ‰) of fatty acids and isoprenoid hydrocarbons from different experiments compared
to values obtained from the original sediment in the methanogenic zone.

| | | | Fatty acids | | Hydrocarbons | |
|---|---|---|---|---|---|---|
| Description | Temperature (°C) | Sampling (days) | $C_{16:1\omega9/8/7}$ | $C_{16:1\omega5}$ | Phytane | Biphytane |
| Pre-incubated slurry +$^{13}$CH$_4$+hematite | 20 | 411 | -40 | -43 | -17 | -23 |
| Pre-incubated slurry +$^{13}$CH$_4$ (bottle A) | 20 | 411 | -40 | -43 | -13 | -24 |
| Pre-incubated slurry +$^{13}$CH$_4$ (bottle B) | 20 | 1227 | -36 | -41 | -5 | -38 |
| [a] Fresh batch experiment+$^{13}$CH$_4$+hematite | 20 | 470 | 610 | 1600 | -14 | -28 |
| Semi-bioreactor+$^{13}$CH$_4$+hematite | 16 | 382 | n.d. | n.d. | n.d. | n.d. |
| Original sediment (28-30 cm) | 14 | | -44 | -51 | -32 | -33 |

[a] Bar-Or et al., 2017
n.d. – Not detected


**4. Discussion**
**4.1 Anaerobic oxidation of methane in the methanogenic sediment incubation experiments**
The *in-situ* geochemical and microbial diversity profiles (Bar-Or et al., 2015) and the geochemical
(Sivan et al., 2011; Bar-Or et al., 2017; Fig. 3) and metagenomic (Elul et al., 2021) analyses of batch
incubations with fresh sediments provided strong support for the occurrence of Fe-AOM in sediments
of the methanogenic zone below 20 cm. Such profiles and alongside incubations showed an unexpected
presence of aerobic bacterial methanotrophs together with anaerobic microorganisms, such as
methanogens and iron reducers (Adler et al., 2011; Sivan et al., 2011; Bar-Or et al., 2015; Bar-Or et al.,
2017; Elul et al., 2021). These findings suggested that both *mcr* gene-bearing archaea and aerobic
bacterial methanotrophs mediate methane oxidation. In the current study, we have supportive evidence
of considerable AOM in the long-term incubations, even after the two treatment stages and considering
the low abundance of the microbial populations.
The data from the second stage incubations show a similar increasing trend in the $\delta^{13}C_{DIC}$ values of both
natural (methane-only) and the hematite amended treatments (Fig. 3). This deviates from our
observations during experiments B and C with fresh sediment, wherein higher $\delta^{13}C_{DIC}$ values were

obtained after the addition of hematite than in the methane-only treatment (Fig. 3 and Bar-Or et al.
(2017)). This was particularly dramatic in the batch slurries (experiment C), but it was also observed in
the semi-continuous bioreactor (experiment B). We assume that the observed difference in the
bioreactors would have been more pronounced if methane concentrations had been higher, but it is still
a relevant finding. We also note that the difference between the bioreactors results may also be due to
the fact that each bioreactor community developed separately. The results of the type A experiments
(compared to those of types B and C) suggest that either hematite lacks the potential to stimulate the
AOM activity during the two-stage experiments or that there is enough natural Fe(III) in the sediments
to sustain the maximum potential of Fe-AOM. Below we characterize the AOM process in the long-
term, two-stage incubation experiments.
**4.2 Potential electron acceptors for AOM in the long-term two-stage incubation experiments**
4.2.1 Metal oxides as electron acceptors
Measurements of $\delta^{13}C_{DIC}$ show that the additions of magnetite, amorphous iron, clays and manganese
oxide in the second stage incubations resulted in a less pronounced increase in the $\delta^{13}C_{DIC}$ values
compared to those of the methane-only controls (Fig. 4). A possible explanation for the latter may be
that these metal oxides inhibit AOM, either directly or via a preference for organoclastic iron reduction
over Fe-AOM, which adds a natural, more negative carbon isotope signal from the organic materials
rather than the heavy carbon from the $^{13}$C-labeled methane. Using mass-balance estimations in the
methane-only and in the amorphous iron treatments and considering the DIC concentrations and $\delta^{13}C_{DIC}$
values of the methane-only treatments at the beginning of the experiment (6 mM and 60‰,
respectively) and the values at the end (6.5 mM and 360‰, respectively), about 0.5 mM of the DIC was
added by the AOM of methane with $\delta^{13}$C of ~4000‰. The DIC and $\delta^{13}C_{DIC}$ values of the amorphous
iron treatment at the beginning of the experiment were 5.4 mM and 60‰, respectively, and by the end
were 6.1 mM and 120‰, respectively. Assuming the same $\delta^{13}$C of the added methane of 4000‰ and a
$\delta^{13}C_{TOC}$ of $-30$‰ (Sivan et al., 2011), 0.1 mM of the DIC should derive from AOM and 0.6 mM from
organoclastic metabolism. This means that adding amorphous iron to the system encouraged iron
reduction that was coupled to the oxidation of organic compounds other than methane. Intrinsic
microbes, particularly the commonly detected ex-deltaproteobacterial lineages such as Geobacterales,
may catalyze Fe(III) metal reduction, regardless of AOM (Xu et al., 2021). Manganese oxides are found
in very low abundance in Lake Kinneret sediments (0.1 %, Table S1 and Sivan et al., 2011). Thus, their
role in metal-AOM is likely minimal.
4.2.2 Sulfate as an electron acceptor
Sulfate concentrations in the methanogenic Lake Kinneret sediments have been below the detection
limit in years past, similar to their representation in the natural sediments we used for the incubations
(< 5 μM, Bar-Or et al., 2015; Elul et al., 2021). Sulfide concentrations have also been reported to be
minor (< 0.3 μM, Sivan et al., 2011). However, sulfate could theoretically still be a short-lived
intermediate for the AOM process, as pyrite and FeS precipitate in the top sediments, and cryptic
cycling via pyrite or FeS may replenish the sulfate, thus rendering it available for AOM (Bottrell et al.,
2000). The addition of Na-molybdate to the second stage slurries, including those amended with and
without magnetite, did not change the $\delta^{13}C_{DIC}$ dynamics, which remained similar to those from before
the addition of the inhibitor (Fig. 4A). This finding is in line with that in fresh batch sediment slurries
(Bar-Or et al., 2017) and suggests that sulfate is not a potent electron acceptor for AOM in this
environment. Furthermore, although sulfate-reducing bacteria were abundant, none of the reducers
belonged to the known clades of ANME-2d partners, which were connected previously to the Fe-S-$CH_4$
coupled AOM (Su et al., 2020; Mostovaya et al., 2021).
4.2.3 Nitrogen species as electron acceptors
Nitrate and nitrite concentrations are also undetectable in the porewater of Lake Kinneret sediments
(Nüsslein et al., 2001; Sivan et al., 2011), but again may appear as short-lived intermediate products of
ammonium oxidation that is coupled to iron reduction  (Tan et al., 2021; Ding et al., 2014; Shrestha et
al., 2009; Clement et al., 2005). We thus assessed the roles of nitrate and nitrite as electron acceptors in
the two-stage slurries. Our results indicate that the addition of nitrate did not promote AOM, likely due
to the absence of ANME-2d, which is known to use nitrate (Arshad et al., 2015; Haroon et al., 2013).
In the case of nitrite, even low concentrations appeared to delay the increase in $\delta^{13}C_{DIC}$ values,
suggesting that organoclastic denitrification outcompetes AOM, and despite the occurrence of
Methylomirabilia, the role of nitrite-AOM is not prominent in the two-stage incubations (Figs. 4C, D).
4.2.4 Humic substances as electron acceptors
Humic substances may promote AOM by continuously shuttling electrons to metal oxides (Valenzuela
et al., 2019). Though humic substances were not measured directly in Lake Kinneret sediments, the
DOC concentrations in the methanogenic depth porewater were previously found to be high (~1.5 mM,
Adler et al., 2011), suggesting that they may play a role in AOM. Compared to the methane-only
treatments, the treatment with the synthetic humic analog AQDS caused an increase in dissolved Fe(II)
concentrations, but it did not cause $^{13}$C-DIC enrichment. This may be explained by the behavior of
AQDS as a strong electron shuttle in organoclastic iron reduction (Lovely et al., 1996), which produces
isotopically more negative carbon that masks the AOM signal (Fig. 4E, Fig. S3). Yet, as was done by
Valenzuela et al. (2017), the addition of natural humic substances did promote AOM, compared to the
rest of the electron acceptors tested, and may thus support AOM (Fig. 4B). In our incubations, the
natural humic substances promoted first the oxidation of organic matter by iron reduction, probably by
shuttling electrons from the broad spectrum of organic compounds to natural iron oxides (Figs. 4B and
5). When the availability of the iron oxides or the organic matter decreased, humic substances likely
took over to facilitate the AOM (Fig. 4B).
Overall, the results of our long-term two-stage experiments indicate that sulfate, nitrate, nitrite and
manganese oxides do not support AOM in the methanogenic sediments of Lake Kinneret. The candidate
electron acceptors for AOM in the long-term experiments are natural humic substances and/or naturally
abundant iron minerals. Future experiments can simulate iron limitation and the involvement of iron
oxides in the AOM by removing natural iron oxides from the sediments.
**4.3 Main microbial players in the long-term two-stage slurries**
Methane oxidation in the pre-incubated Lake Kinneret sediments is likely mediated by either ANMEs
or methanogens, as the addition of BES and acetylene immediately stopped the AOM (Fig. 6) similar
to the results of the killed bottles and the BES treatment in the fresh batch experiment (Bar-Or et a.,
2017). Apart from methane-metabolizing, acetylene can inhibit nitrogen cycling, which results in
ethylene production (Oremland and Capone, 1988). This was not the case in our incubations, as no
ethylene was produced. The increase in $\delta^{13}C$ values in phytane and biphytane (Table 3) also indicates
the presence of active archaeal methanogens or ANMEs (Wegener et al., 2008; Kellermann et al., 2012;
Kurth et al., 2019).
Using the isotopic compositions of specific lipids and metagenomics, we identified a considerable
abundance of aerobic methanotrophs and methylotrophs in the fresh sediments, but not in the long-term
slurries (Table 3, Fig. 7). In the natural sediments, micro levels (nano molar) of oxygen could be trapped
in clays and slowly released to the porewater (Wang et al., 2018). However, if such micro levels of
oxygen still existed during the time of the pre-incubation, they were probably already exhausted.
Indeed, the results of our specific lipids and metagenomics analyses suggest that the aerobic
methanotrophs lineages play only a minor role in the long-term slurries, probably due to complete
depletion of the oxygen. The metagenomic data (Fig. 7, Supplementary coverage table) also indicate
that Bathyarchaeia, which may be involved in methane metabolism (Evens et al., 2015), were enriched
in the bioreactor incubations, yet their role in Lake Kinneret AOM remains to be evaluated. We also
observed changes in the abundance of bacterial degraders of organic matter and necromass: for example,
GIF9 Dehalococcoidia, which can metabolize complex organic materials under methanogenic
conditions (Cheng et al., 2019; Hug et al., 2013), were most abundant in the long-term incubations (Fig.
7, Supplementary coverage table). Though ANME-1 are likely mediators of AOM in these sediments,
methane oxidation via reverse methanogenesis is feasible for some methanogens in Lake Kinneret
sediments (Elul et al., 2021).
**4.4 Mechanism of methane oxidation in the long-term two-stage incubations**
Our results indicate net methanogenesis in the two-stage incubation experiments with an average rate
of 25 nmol $g^{-1}$ DW $day^{-1}$ (Fig. 1 and Table S2), which are similar to those from fresh incubation
experiments (Bar-Or et al., 2017). This is despite the overall trend of increasing $\delta^{13}C_{DIC}$ values, a result
representing potential methane turnover (Figs. 3 and 4). A likely explanation for the presence of both
signals is an interplay between methane production and oxidation, which is possibly triggered by
reversal of the methanogenesis pathway in bonafide ANMEs or certain methanogens (Hallam et al.,
2004; Timmers et al., 2017). Due to the overall production of methane and the lack of intense
stimulation of AOM by any electron acceptor added, the increase in $\delta^{13}C_{DIC}$ values could theoretically
result from the occurrence of carbon back flux during methanogenesis, which is feasible in
environments that are close to thermodynamic equilibrium (Gropp et al., 2021). To test this, we used
DIC mass balance calculations to determine the strength of back flux in our incubations. Based on
equations 1 and 2, the observed level of $^{13}$C-enrichment indicates that 3-8% of the $^{13}$C-methane should
be converted into DIC. These estimates are orders of magnitude higher than the previously reported
values of 0.001-0.3% for methanogenesis back flux in cultures (Zehnder and Brock, 1979; Moran et al.,
2005), but they are in the same range as the back flux of 3.2 to 5.5% observed in ANME-enrichment
cultures (Holler et al., 2011). For the latter, however, modeling approaches from AOM-dominated
marine sediment samples and associated ANME enrichment cultures indicated the absence of net
methanogenesis (Yoshinaga et al., 2014; Chuang et al., 2019; Meister et al., 2019; Wegener et al., 2021).
Thus, it seems unlikely that back flux alone can account for the methane-to-DIC conversion in Lake
Kinneret sediments. Moreover, the occurrence of back flux alone in marine methanogenic sediments
with similar net methanogenesis rates and abundant methane-metabolizing archaea did not yield
considerable $^{13}$C-enrichment in the DIC pool following sediment incubations (Sela-Adler et al., 2015;
Amiel, 2018; Vigderovich et al., 2019; Yorshansky, 2019) (Table S3). It is, therefore, less likely that
the observed DIC values in our study were sustained by methanogenesis back flux alone (without an
external electron acceptor) than by active AOM, which, in this case, is probably performed by ANME-
1 or by methanogens, with the latter performing reverse methanogenesis to some extent.
**Conclusions**
Previous results of geochemical and microbial profiles as well as incubations with fresh sediments from
Lake Kinneret constitute evidence of the occurrence of Fe-AOM in the methanogenic zone. The process
is performed by anaerobic archaeal methanogens and aerobic bacterial methanotrophs, which remove
about 10-15% of the methane produced in the lake's sediment. In the current study, we found that after
two incubation stages and intensive purging for a prolonged duration, AOM was still significant,
consuming 3-8% of the methane produced. However, the abundance of aerobic methanotrophs
decreased and anaerobic archaea (ANME-1 or specific methanogens) appeared to be solely responsible
for methane turnover. AOM could be a result of carbon back flux, as the methanogenic/AOM pathway
is reversible, however, the high $\delta^{13}C_{DIC}$ signal points to a metabolic reaction. Terminal electron
acceptors or electron shuttles stimulating Fe-AOM are either hematite and/or humic substances. The
role of the aerobic methanotrophs of the order *Methylococcales*, which were found in the freshly
collected sediment experiments, remains to be examined.
**Competing interests.** The authors declare that they have no conflict of interest.
**Acknowledgments**
We would like to thank B. Sulimani and O. Tzabari from the Yigal Allon Kinneret Limnological
Laboratory for their onboard technical assistance. We thank all of O. Sivan's lab members for their help
during sampling and especially heartfelt thanks to N. Lotem for the invaluable assistance with the mass
balance calculations and the fruitful discussions and E. Eliani-Russak for her technical assistance. Many
thanks to K. Hachmann from M. Elvert's lab for his help during lipid analysis and to J. Gropp for
insightful discussions about the back flux. This work was supported by ERC consolidator (818450) and
Israel Science Foundation (857-2016) grants awarded to O. Sivan. Funding for M. Elvert was provided
by the Deutsche Forschungsgemeinschaft (DFG) under Germany's Excellence Initiative/Strategy
through the Clusters of Excellence EXC 309 'The Ocean in the Earth System' (project no. 49926684)
and EXC 2077 'The Ocean Floor—Earth's Uncharted Interface' (project no. 390741601). Funding for
M. Rubin-Blum was provided by the Israel Science Foundation (913/19), the U.S.-Israel Binational
Science Foundation (2019055) and the Israel Ministry of Science and Technology (1126). H.
Vigderovich was supported by a student fellowship from the Israel Water Authority.

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
