# Peer review of "Long-term incubations provide insight into the mechanisms of anaerobic"

_Biogeosciences, 2021_

## Author Comment (AC1)

Response to referee #1 (our answers in blue):

General:

I think this manuscript would benefit from some more proofreading by the more experienced authors. It could use improvement on the structure and the writing, to improve the flow and make it more condensed. Please also pay attention to the switching between different tenses, and to improve the clarity of the methods section. Many different experiments have been performed in this study, which is wonderful. It makes it, however, difficult for the reader to keep an overview. Please structure the manuscript in a way that provides the necessary overview and clarity. Present the results in a structured way in the methods section, and don't be tempted to already interpret them – this belongs to the discussion. Also prevent the use of language that is either too strong (This means..), or is not specific enough (warm, very few etc.) Overall, I think the experiments are cool and valuable, but improvement is needed to bring this across to the reader.

We would like to thank this reviewer for the positive and constructive review. We have carefully responded to each comment/suggestion and did our best to improve the manuscript accordingly. We clarified and simplified the paper all along: we re-wrote the methods section to make it easier for the reader to follow, we also added experimental protocols to the supplementary material. We moved discussion parts from the methods and the results to the discussion section. The microbial data was discussed more thoroughly and a principal component analysis biplot was added. All co-authors edited the letter and the manuscript, with additional English editing by a native speaker collaborator. We believe that the paper flows better now, making it easier for the reader.

Abstract

Introduction about sediments is too long. Could skip most of it, one or two sentences is enough. Instead, tell us more about the two stages of incubations and 13C additions, multiple TEA and inhibitors. What did you use, what were the aims? If you don't want to stress these, give less detail, now it creates more questions than answers.

We shortened the introduction and added more details on the experimental design, the different treatments (electron acceptors and inhibitors) and their aims.

25-27. This sentence is a bit clunky, with the two words for the same process (oxidation and AOM). Also, here you name it methanic sediments while these were the incubations/reactors right?

Thank you for this comment, we adjusted the sentence. The word "oxidation" was changed to "AOM" and "methanic sediments" to "incubations".

The abstract could use re-structuring, please have in mind what are the most important messages you want to convey, stress those and don't give too much details about other things. It could also be nice to give one or two sentences at the end that place your results into a broader context.

We accept this comment. Unnecessary information has been removed and a sentence has been added to emphasize our results in a broader context.

Keywords: I would add mcr and methanotrophs

Thank you, we have added these words.

General textual: Methanic is not a word that is commonly used I think. Methanogenic is the more general term, at least, I think that is what you mean? But this is personal preference, to choose what you want to use.

We switched to the term to "methanogenic" as the reviewer suggested. It should be noted that some methane researchers use the term "methanic" as more general term that refers to an environment where methane is present but not necessarily produced locally. As here we are certain that methane is produced *in situ*, the term "methanogenic" indeed is more appropriate.

Methods

98. If you want to say it's warm, give a temperature. The name "warm monomictic lake" is well established in limnology and refers to lakes that never freeze, so we kept this use.

99. Similar to what? Similar to previous studies that were mentioned in the text. The text has been modified accordingly.

100. Are there methane profiles? Yes, and the relevant information has been added.

101. You have not mentioned the central lake or station A yet. The reviewer is correct, and this information has been added.

102. which leaves = leaving. Corrected

103. Did they receive new methane after that? Methane was added after $N_2$ flushing.

104. This sentence is weird, 'in case of' is not fitting. We have rephrased the sentence.

105. This seems more like discussion or results, not methods ('the variations…'). This sentence was moved to the discussion.

106. The black coffee comes out of nowhere and the explanation about why only 1 replicate is not fitting. The black coffee treatment was removed from the manuscript.

This whole paragraph is chaotic, try to restructure to make it a bit more schematic and easier to follow, to help the reader understand. We changed the methods section to be clearer. We describe the sediment collection, then the set-up of the pre-incubation slurries and then the two-stage incubation experiments in general. The experiments table was moved to the main text, and protocols for the experiments were added to the supplementary material.

Do you mean real porewater every time you write porewater, or an artificial substitute? It seems like a lot of porewater to extract, which is possible I guess, but I'm just not sure and curious!

Indeed, we used real porewater for preparing the slurries in the experiments (a lot of work….). We collected many sediment cores in each campaign and extracted porewater from the bottom sediments (>20 cm) to mimic *in situ* conditions. We further clarified this point in the revised text.

192. Don't switch between past and present tense within a paragraph. The tenses were corrected.

249 I don't think this paragraph is necessary. We cut part of it and added a short overview of the results, as was suggested by another reviewer.

255. Can you start with simply describing your results? You dive in deeply directly, it would be nice as a reader to get a bit of a gentle overview first, of what you measured and

what that showed, to start with. We accept this comment. In the revised version, we start with an overview of the concept and what we measured.

256. No need to note that here. The sentence was moved to the methods section.

257. This was not subsequent but different experiments, right? The word first suggests otherwise. We meant that metals were the first type of electron acceptors tested. The word "first" was changed.

258. Discussion, not results. Stick to just listing the results, so the values that you measured and their patterns, here. We removed all discussion from results. We only kept the indication for AOM by using the transformation of $^{13}$C methane to $^{13}$C-DIC.

Fig. 2. What is the difference between the colors of the pre-incubated experiment? The legend calls them the same.

The legend of figure 2 has been modified to be clearer regarding the experiments. It now includes numbering of the experiments which correspond to the numbering in the experiments detail table. Each color is an experiment, and all of them have the similar treatments of "only methane" and "hematite". In this graph we wanted to show that as opposed to what was shown previously with freshly collected sediments, we do not see a clear difference between the two mentioned treatments in the two-stage experiments.

Fig. 3. The text is too small and therefore hard to read. Why don't you merge the replicates of each treatment into one line with error bars? They seem to nicely follow the same trends. Also, it would be nice to have the same y-axis and x-axis for easy comparison between the treatments.

We merged the replicates as suggested and changed the y and the x axis.

Fig. 4. Similar to Fig 3: please merge the lines of the replicates.

Lines were merged.

Table 1. The names of the treatments could be improved. What is a typical fresh sediment bottle?

The names have been changed. The fresh sediment bottle is the result from the freshly collected sediment slurry experiment, and its title was changed as well.

I'd be happy to provide more comments on a next version of the manuscript.

Thank you!

---

## Author Comment (AC2)

Response to referee #2 (our answers in blue):

The paper by Vigderovich et al. investigated the pathways of anaerobic methane oxidation in Lake Kinneret sediments by a combination of incubation techniques, lipid and metagenomic analyses.

The authors performed a series of long-term incubations in bottles and semi-bioreactors with an array of added potential electron acceptors and inhibitors for specific metabolic processes in order to track down the dominant processes responsible for AOM. The results obtained from this study were interpreted in combination with results obtained from previous studies on these sediments. All in one, the experimental design was thorough and the use of combinations of electron acceptors/inhibitors feasible for interpretation of possible AOM pathways in these sediments.

We would like to thank the reviewer for the constructive review and the approval of our experimental design and selection of e-acceptors. Our main goal was to cover all potential electron acceptors and scenarios with many experiments in a comprehensive way.

The paper is mostly focusing on presentation and interpretation of geochemical data. The authors did perform taxonomic read and metagenomic analyses from several incubations and incubation time points, but I miss the presentation of these results in the paper. The results are briefly mentioned, but I would prefer to see a visual representation of DNA-based results in a separate section in the 'Results' section and a more thorough integration with lipid and geochemical analyses.

We agree with the reviewer's comment that the previous version focused almost entirely on the presentation and interpretation of geochemical data and that we should put more emphasis on the molecular-biological results. As suggested by the reviewer, we added a microbial section with visual representation of DNA-based results (in addition to the table) and integrated this with the other analyses.

It was also a little confusing to see a 'black coffee' treatment, as there was no introduction or reasoning why this rather unusual substrate was used for AOM incubations. Also there was no detailed protocol on how this treatment was prepared (what fraction, concentration etc). Every treatment should be reproducible from the information provided by the paper, but here any details are lacking. So I would suggest to either remove this from the paper completely or to describe the treatments and reasoning thoroughly.

The "black coffee" means coffee grinds as an organic source. However, as this treatment seemed confusing to all reviewers, we decided to remove it from the paper.

In general, the results presented in this paper are interesting and will benefit the scientific community investigating AOM in natural sediments. The paper will benefit from a more clear structure and better visual presentation of results.

As mentioned above in the reply to the first reviewer, we improved the structure of the manuscript, clarified and simplified the paper, all co-authors edited the manuscript and we also added final English editing.

---

## Author Comment (AC3)

Response to referee #3:

This work by Vigderovich et al., investigated the key microbial players and electron acceptors that support anaerobic oxidation of methane, methanogenesis and possibly a sulfur cycle in in the top 20 cm of sediments collected from a lake in Northern Israel. They used a variety of sediment slurry incubations amended variety of electron acceptors, electron acceptor analogs and inhibitors along with $^{13}$C labeled methane and tracked the buildup of $^{13}$C-DIC over time. Their results indicate that there is methane oxidation occurring by aerobic methane oxidizing bacteria and anaerobic methane oxidizing archaea (ANME) possibly with oxygen or iron oxides. However, later in the experiments the data suggest that humic substances are the most likely culprit for the turnover of $^{13}$C-methane. Their metagenomic data suggest the presence of methanogens, aerobic methanotrophic bacteria and anaerobic methanotrophic archaea and suggested that there is an interplay between the groups that cycle the carbon in their experiments.

Generally, I find the data to be very interesting and does fit well within the scope of Biogeosciences and should be eventually accepted with major revisions. My main critiques for the manuscript are in the clarity, flow in all sections and a few discussion inquiries I would like to address in this review.

We would like to thank the reviewer for the positive and very instructive review. We realized based on all reviews that the manuscript was not perfectly presented. As mentioned above, we clarified and simplified the paper throughout; we re-wrote and organized the methods section and added experimental protocols to the supplementary material. We moved discussions from the results to the discussion section. The microbial data was discussed more thoroughly and a principal component analysis biplot was added. All co-authors edited the manuscript, and we also edited the English. We believe that the paper flows better now and is much clearer for the readers. We also accepted all other comments, which we believe improved the revised version.

General comments:

The introduction lacks a clear identification of the gap in the knowledge to which scientific questions are based on. The question and hypothesis in L84-86 is rather vague and could be clearer.

We agree, and we clarified in the introduction the gap of knowledge, the aims and the hypothesis, as explained in the response to the first reviewer.

The methods unfortunately are riddled with syntax errors and missing study site and methodological details that need to be clarified. This is particularly important as methods sections should be written such that anyone could reproduce the experiment. This review cannot identify and fix all of them but will provide examples of some of the most severe below.

We totally agree and have supplied more details on the methods such that every treatment should be reproducible and clear. We accepted all comments below and edited the methods section accordingly. We also added the protocol for each experiment in the supplementary information.

The beginning portions of the results section sound more like a discussion. There is more space being used to repeat the experiments and experimental setup in this section than simply reporting the data from the experiments. The results section would be stronger if the authors would report the data without method explanations or with interpretation. Further into

the section there are more numbers that are reported but there are other sections that read crudely. Consider having introductory and conclusion sentences and sub-headers (applies to other sections) to better separate sections which will help with flow.

We accept this comment. In the revised version, we improved the results chapter by starting with a short introductory and overview of the results (as also requested by the first reviewer), by reporting the data with minimum interpretation but with some conclusion sentences to better separate sections and to guide the reader.

Figures 2 and 3 could be organized better. There is a lot of data and the scales do not match which could be misleading at a first glance. Furthermore, there is some data in the supplementary material that belongs in the main text. For example, the authors suggest that aerobic methanotrophic bacteria play a role in the overall AOM process. Their geochemical data do not definitively track aerobic methanotrophic bacteria activity (not sure you even could) but the molecular data do indicate their presence.

We reorganized the figures and added the microbial data and its indications to the main text, as was similarly requested by the second reviewer.

In the discussion are a lot of interpretations and claims for which are speculative and do not offer enough literature examples that support the interpretations. For example, in L364: I don't think that the addition of iron-oxides generally inhibits AOM according to your data. Where in the literature do you find an example of iron-oxides acting as an inhibitor to AOM? Yes, there seems to be less buildup of your 13C-DIC but the system still observes a buildup of 13C-DIC after ~450 days in figure 2 and in figure 3A after 450 days. If it was truly an inhibitor then the 13C-DIC would be identical to the killed control trend throughout the whole experiment, just like the BES trend in Figure 4 where you know BES is inhibiting activity. But Figure 3A lacks a killed control for hematite so how does one know that the addition of hematite is truly an inhibitor? Instead, you observe the trend to become slightly depleted in 13C before the full spike, then see a rebound in the hematite trend back to levels closer to the beginning of the experiment, in which case how do you explain that? In addition, according to Fig 3A, all 4 amendment experiments had decreasing 13C-DIC which leads me to believe that there might be something else at play perhaps the organoclastic iron reduction as the authors mentioned or something in the experimental setup that is causing all of your replicates to all have decreasing 13C-DIC.

It seems that our statements were not as accurate as we thought. We thus clarified them in the revised version: We are discussing "inhibition" here not as direct damage to the pathway of AOM, but rather making another process more favorable than the AOM. Perhaps a more accurate phrasing would be "an inhibition in the AOM signal". We are considering the $^{13}$C-enrichment in the DIC in the treatments without adding an electron acceptor as the baseline of these slurries. The methane oxidized in those treatments should be about the same in every other bottle in the experiment unless the addition of the electron acceptor discouraged the AOM (except for the killed controls). If the $\delta^{13}C_{DIC}$ values are lower than the "baseline" when an electron acceptor is added, it could be from the following reasons: 1. there is another microbial process converting organic carbon to inorganic carbon in addition to the methane oxidation. This means that the DIC pool (which consists of all the products of organic matter oxidation in the slurry) in those treatments is diluted with $^{12}$C compared to the treatments without an additional electron acceptor, which comes from another source other than methane (since the same amount of labeled methane was added to all the experiment bottles in each experiment). In most experiments, we only added $^{13}$C-labeled methane to begin with so a decrease in isotopic value due to methane oxidation is unlikely. 2. It directly interferes with the microbial pathway of the AOM. As the reviewer mentioned, there is still an increase of $\delta^{13}$C-DIC in the incubations containing different electron acceptors, and therefore

we believe that the former option is the correct one. When we added an electron acceptor and saw that the $\delta^{13}C_{DIC}$ values were lower than without it, we concluded that either there is an inhibition of the methane oxidation (less methane is being oxidized now) or that there is much more oxidation of $^{12}C$- organic carbon than without the electron acceptor addition, which results in a dilution of the final DIC pool.

Specific inline comments and edits:

L44: AOM coupled to other electron acceptors is not a theory anymore. The word "theoretically" was deleted.

L49-50: Consider adding equations of the AOM reactions. We did not add the equations as they can be found in the literature, however can add them to the supplementary if needed.

L47-L54: Consider joining L47 into next paragraph at L50. The lines were joined.

L74-75: Move citation to the end of sentence. The citation was moved.

L111-112: "1) Two stage slurry incubations with 1:1 sediment - pore water ratio for three months, followed by a 1:3 ratio and the addition of different manipulations for up to 18 months." This sentence is far too long and could be split up to be clearer about the analysis. The sentence was split.

L113-114: "Semi-continuous bioreactor experiments with freshly collected methanic sediments and porewater with 1:4 ratio, where porewater was exchanged regularly." Syntax error, do you mean 1:4 sediment to porewater ratio or do you mean the porewater has a 1:4 ratio, in which case what is the 1:4 ratio in the porewater? Same can be said for L115-117. We meant 1:4 sediment to porewater ratio. This was clarified in the text.

L97: The methods would also greatly benefit with a couple sentences that explain how the sediments were retrieved from the Lake (i.e. ship/small boat, instruments (mulitcorer, pushcores, gravity cores etc…). The text was edited to include information about how the sediments were retrieved from the lake.

L98-106: This section is rather vague, I think it would be stronger with more details about the lake such as; approximate temperature and size and perhaps a nearby city for reference. A map would also make this section stronger. The details were added to the text and a map was added to the supplementary.

L109: Here you mention you are going to assess the different electron acceptors for AOM. It may be wise to have a sentence somewhere either in the methods or in the intro that you are lumping all methane oxidation by archaea (ANME's) and bacteria (Methylococcales). Many in the community may just be thinking AOM process is being conducted by the ANME's but it appears (not clear) you are referring to both, correct? Yes, we wanted to see if the aerobic methanotrophs are still active and involved in the two-stage incubations. We thank the reviewer for pointing this out. A sentence was added to the introduction clarifying that we were investigating whether both archaea and bacteria are still responsible for methane oxidation in the long-term anaerobic incubations.

L118: This section would be also stronger with one sentence explaining what the purpose of the two-stage incubation is. The purpose of the two-stage incubation was clarified in the text. It was done in order to enrich the microbial population that is involved in the methane oxidation process in the natural Lake Kinneret sediments fast by slight dilution with

porewater, and then to explore the long-term incubation effect on AOM process with larger dilution. Sediments were therefore pre-incubated in a 1:1 sediment-porewater ratio with total of 20% methane in the headspace (18% natural methane (98.9 % 12C, 1.1 % 13C) and 2% 13C- labeled methane (99% 13C)). When we observed an AOM signal (13C-DIC enrichment) we diluted the slurry once more (1:3 sediment-porewater ratio), added different treatments and started the experiment.

L119-120: Add more information of where you sampled perhaps on a map. I do not know where "Station A" is. Additionally, do you know precisely when the sediments were collected between 2017 and 2019? How long did it take for the sediments to be processed into sediment slurries? Were they stored and reactivated? If sediment samples that were collected in 2017 and processed in 2019 how do you know those samples are still viable for this study? Station A is located at the center of the lake, which is also the deepest point of the lake (water depth of 42 m), and it is shown on the added map in the supplementary. From 2017 until 2019, we undertook multiple cruises to Station A to collect the sediments for our slurry experiments using a gravity core with Perspex liners (60 cm long, inner diameter 5 cm). The cores were kept at $4°C$ until further processing (up to 48 hours). Targeting solely the zone below 15 cm, sediments were transferred into 250 ml bottles, then diluted with porewater at a 1:1 ratio. The latter was extracted from the same depth using parallel cores. After three months of incubation, the slurry was divided into new bottles and diluted again to reach 1:3 sediment to porewater ratio. To these bottles, different electron acceptors/inhibitors were added. The sediments in all the experiments were set up immediately (within a day of collection), but they were spread along the two years. The methods section was edited to include and clarify these details.

L120: What was the container that the sediments were pooled into? The sediments were collected using a gravity corer with 50 cm Perspex cores and sediments from the methanogenic depth (below 20 cm) were incubated in 250 ml bottles.

L121: Please add the speed (rcf x g) for the centrifugation. The speed (9300 g) for the centrifugation was added.

L133-134: What do you mean "already running experiment" is this separate from the two-stage experiment? If so, this was never introduced. What we meant was that the inhibitor was during the second stage to the specific bottles. This was clarified in the text.

L138-139: Syntax; this sentence makes it seem like there is a separate experiment within one. Was there a reason not to add acetylene in the beginning like BES? The sentence was corrected to clarify that BES and acetylene were added to two different experiments other than the one with the addition of molybdate. Acetylene was not added at the beginning of the experiment because this was the first time we used it, and we wanted to observe first the $^{13}C$ enrichment in the DIC in the specific bottle before the inhibition. Since the addition was relatively easy by injecting gas to the headspace, we decided that this would be the way to see that acetylene really stopped the AOM.

L145-147: It was mentioned that $^{13}C$-label was added after the pre-incubation, please add the time you added the label. The $^{13}C$-labeled methane was added to the bottles immediately when we set the incubations (within 48 hr from sampling).

L150: Never begin a sentence with numerical, instead spell out two. Thanks for this comment. It was corrected

L150- What method and instrument was used to measure the Fe(II)? To measure Fe(II) we fixed the filtered porewater immediately after sampling with Ferrozine (a chelator that creates

a strong complex with Fe(II)) according to Stooky (1970). The absorbance of the complex is measured by spectrophotometer at 562 nm wavelength. It is described and clarified better in the analytical methods (the former porewater analyses section).

L152- How was the methane measured? The methane was measured from the headspace after shaking (to allow the methane to transfer from the porewater to the headspace) on a focus gas chromatograph (GC) equipped with a flame ionization detector (FID). It is described in the analytical methods section (former porewater analyses section).

L153- Is there an equation used to calculate methane? Methane amount in the headspace was measured using a calibration curve. The concentrations in the porewater were calculated using the known bottle volume and slurry volume. We assume that every change in the concentration is due to methanotrophs/methanogens activity in the slurry. This was clarified better in the text.

L154-155: This is the first time the "Black Coffee experiments" was introduced. Please add why you included this. The black coffee means coffee grinds as an organic source, and was added to test whether another type of complex organic compound influences the oxidation of methane. However, we removed this treatment from the manuscript, as the other reviewers suggested.

L154: Is there a reason why there is inconsistencies between duplicates or triplicate samples (i.e. not enough sample or prioritized sample for certain treatments of interest?). Please elaborate. Yes, due to the limited availability of porewater and sediment, samples were prioritized, and we needed to dictate how many replicates would be set up.

L156-157: This sentence is contradicting and with parentheticals is incomplete. The sentence was rewritten.

L156-157: This is the first mention of any killed controls in this part of the experiments. How many were there? How were they prepared? (etc). Killed controls (triplicates or duplicates) were part of each set of sediments. Killed control bottles with sediment and porewater were autoclaved twice after they were flushed with $N_2$ together with the rest of the experiment bottles. After they cooled down, the relevant additions (i.e. electron acceptors and $^{13}C$-labeled methane) were added. This was clarified in the text. Given the quantity of experiments presented, a detailed description of how many replicates for each treatment was included in table S2 in the supplementary, including the killed controls.

L158- What is so special about the lake in Alaska that humic substances had to be extracted from. Why not get them from Lake Kinneret? We are working on extracting humic subtances from Lake Kinneret, which would be the best option. However, it is not a trivial procedure. We decided, therefore, to add natural humic substances that we already had available from a lake in Alaska when we set up that specific experiment. This allowed us to compare our results from the synthesized analogue with a natural one.

L164-165: What was the other bioreactor amended with? Or is it a control? The second bioreactor was set up as a control, without iron oxides. This was clarified in the text.

L165-166: Syntax issues, had to read it several times over to understand that you are trying to describe how 13C methane was added to the headspace free bioreactor. The sentence was changed to: "To dissolve $^{13}C$-labeled methane in the porewater, 15 ml of porewater were replaced with 15 ml of methane gas to produce methane-only headspace for 24 hours."

L170-172: How many total weeks did the bioreactor run for? The bioreactor ran for 67 weeks. The duration of the reactor was added to the text.

L175: This is the first time that a duration of the experiment has been introduced. Consider adding the actual experiment duration somewhere in the method. The duration of the experiments was added to the supplementary information for each experiment.

L175-177: Good introductory sentence. Please move this sentence to the beginning of the section.The sentence was moved as suggested.

L185: Figure 1 caption should be moved into the text. Particularly you did not describe how you set up the third experiment till the caption. Details from Figure caption 1 were moved to the text. The third experiment was set up and described in detail in the Bar-Or et al., 2017 paper. However, we added also details in the text.

L192: What kind of autosampler? Was this done at the home lab? Yes, this was done at Sivan's lab at Ben Gurion University. The autosampler is a headspace autosampler (CTC analytics. Type PC PAL) which is connected to the IRMS.

L197-199: should be moved to L150.We preferred to keep all the analytical methods in the same section.

L199-202: Which bottle is now being sampled? This is a new section and don't know which experiment is being sampled. The sentence would be stronger if you indicate the reason why you track methane and ethylene (i.e. tracking methanogenesis and acetylene turnover). Methane was measured in the experiment without any electron acceptor in order to assess the methanogenesis rate in the natural sediments that went through the two-stage incubation. This was clarified in the text.

L207 - 208: You list the same variable "x" as two different parameters. This was corrected.

L216: Please indicate which set of experiments the samples come from. The text now specifies from which experiments the samples originated.

L254: I think it would be better to break 3.1 into sub sections to have better flow. It is difficult for the brain to switch between experimental setups. We agree with the reviewer. The section has been divided into sub-sections.

L255: Is the pre-incubated long term experiments the same as the two stage experiments? Please be more consistent with the names of experiments. Yes, they are the same. We made sure that the experiments will be referred as "two-stage experiments" throughout the manuscript.

L255-256: Here is an example of discussion text in the results. How much 13C methane exactly was converted? The sentence was changed, and it now includes the amount of $^{13}$C-methane converted.

L257-258: This sentence sounds like it should be in the discussion. Consider instead to report the actual permit value and leave the microbial population statement for when you report the microbial ecology. The part of the microbial population was deleted, and the rest of the sentence was changed.

L261-263: This sentence is very confusing and how does this relate to the statement you had about AOM in the previous sentence? Please reorganize. This sentence was moved to the methods section.

L260-273: The whole paragraph sounds like it belongs in the discussion. Perhaps move to discussion or add more details about the data. The paragraph was re-edited and now includes specific details about the experimental data without a discussion.

L278-280: Move to methods. The sentence was moved as suggested.

L274: Was sulfate ever measured in your experiments? The natural sediments for these slurries were taken from below 20 cm depth, where sulfate was not observed in depth profiles. Sulfate was also not detected during the fresh slurry experiments (Bar-Or et al. 2017). However, there could be theoretically a cryptic cycle that would produce very low concentrations of sulfate (and consume it fast), and therefore we tested this possibility by inhibiting sulfate reduction with molybdate.

L288: Is the 308 days the end of the experiment? Yes, the duration of the nitrate experiment was 306 days.

L284: End of what? How many days was that? The end of the nitrate experiment (306 days). This was clarified in the text.

L292: What is PCA? Please spell out acronym. What was the result of the AQDS addition? Not clear. Phenazine-1-carboxylic acid (PCA) is an analogue for methanophenazines that are found on the membrane of some methanogens and used to shuttle electrons. The acronym is spelled in the methods section. The addition of AQDS slightly decreased the $\delta^{13}$C-DIC values. This was clarified in the text.

L296 By how much did the Fe(II) and delta 13C increase? Please report. There was an increase of 90 μM in Fe(II), and of ~200‰ in the $\delta^{13}$C-DIC. The text now reads: "The results show that first, the $\delta^{13}C_{DIC}$ values did not change (Fig. 3F), while a steep increase of ~90 μM in their Fe(II) concentrations was observed (Fig. S3). However, after 20 days, the $\delta^{13}C_{DIC}$ values of these slurries started to increase dramatically from 150‰ to 340‰…"

L297: What was the slope? The slope was 2.2 ‰ day$^{-1}$. This was added to the text.

L289-305: The results are very vague and sound more discussion are. Please add in the decreasing and increasing permil and concentrations values for this section. The paragraph was edited to include the concentrations values.

L301: A lot going on in Figure 2 and legend could be better organized. The legend was re-written with new phrasing. The different pre-incubation exp was changed to "two-stage experiment" with numbering of the different experiments (which are now included in the experiments details table) to clarify what is presented.

L311-312: Figure 3. Consider making all y-axis scales the same. We considered this, however in one of the experiments the values reach 2000 permils and it would mask the rest of the trends. We decided to make the y-axis scales of all the graphs the same except for the one with these high values, and we added a line in the caption: "note the different scale of the y-axis in panel E."

Consider moving the Fig 3 F next to Fig 3 A since they both seem to be the experiments that indicate when 13C label was added. Also I do not recall an exact time when the label was added in the methods. We moved the figures as suggested. The labeling time is mentioned in Table S2, which was moved to the main text. It is also clarified in the detailed protocol of each experiment in the supplementary methods section.

Fig 3C: Are the NO3 (grey circles) and the Hematite + NO3 1 mM (green triangles) data on top of each other? Please check your graphs. Yes, they are on top of each other. This was now mentioned in the text.

L319: Please report in text how high of an abundance and which species. We added a supplementary information table that shows the coverage of all the taxon represented by metagenome-assembled genomes.

L320 and 321: Hyphen between "sulfate reducing"

The whole section was re-written, the term "sulfate reducing" was deleted here.

L320-321: Please rewrite sentence. Just report which SRB were present.

The whole section was re-written, and which now includes the SRB that are prominent: Desulfobacterota and Thermodesulfovibrionales (Nitrospirota).

L322: Please report the number of reads to NC-10.

The whole section was re-written, and now includes this data as coverage of the metagenome-assembled genomes.

L339-342: Where are the profiles? Or are you referring to profiles in previous studies? Please clarify.

We are referring to previous studies. This was clarified.

L342-344: Is this really your previous work, or the current work or is this the work from the citations at the end of the sentence? I think you are trying to compare the three different experiments but it makes it sound as if there are three papers in one. The whole paragraph summarizes our group's previous studies on AOM in Lake Kinneret sediments.

L342-344: Is the mechanic zone where Fe-AOM the same sediment regions that you obtained for this study? Yes, they are the same. Previous experiments on Fe-AOM used sediments from the center of the lake and below 20 cm sediment depth. The sediments (and extracted porewater) used for this study were taken from the same spot and depth.

L344-348: Please clarify, I can't tell if you are referring to the current study or other works. We are referring to the current study. This was clarified.

L354-359: Add figure references since you have two figures that compare the three setups. The references were added.

L357-359: But then how do you explain the sharp increase in the 13C DIC in Line 354? As we see the same sharp increase in the incubations without any additions, we cannot state clearly that Fe(III) (as hematite) stimulates AOM as the electron acceptor. It's either that the high amount of Fe(III) in the sediments (3%) is enough to sustain the long term AOM by

reverse methanogenesis or that the long term AOM is stimulated by another electron acceptor.

L364: The methods indicate that the preincubation called for a full methane headspace that was half 12C and half 13C, is it not conceivable that the mass balance would lead to a slight depletion of the 13C in a closed system like this? I would argue that Figure 3A and F shows that before the addition of the 13C label the 13C DIC was similar to the control but after the addition, all trends become heavier. In which case how do you interpret that as an inhibitory response to the addition of label and iron?

Just to clarify, the headspace was never full of methane. In the pre-incubations, 18% was 12C-CH4 and 2% was 13C-CH4. In most of the two-stage experiments, 5% of the headspace was methane (only 13C-CH4). As we mentioned above, we refer to "inhibition" here not as direct damage to the pathway of AOM, but rather making another process more favorable than the AOM. We believe that there is another microbial process converting organic carbon to inorganic carbon in addition to the methane oxidation that is occurring in these slurries. In the experiment presented in figure 3A, the methane added at the start of the experiment was not labeled (a mistake); that is why we do not see an increase, but we see a slight decrease. When $^{13}$C-methane was added, we see an increase in all treatments, but the highest increase was observed in the baseline treatments without any electron acceptor addition.

L367-368: I think it is conceivable to claim that organoclastic iron reduction could dilute the 13C-DIC signal in these experiments, especially over time but do you have any evidence to support that either by isotopioc analysis of organic matter in this study or previous study that could allow you to make some 1$^{st}$order mass balance to explain that? Yes, we have the $\delta^{13}$C value of the organic compounds and did some rough mass balance calculations to show it can lower the signal. This was added to the text.

L372: I agree with your statement of manganese oxide but what do you have to say about the Magnetite additions in Fig. 3A? Those were the most similar to the no electron acceptor control experiment. We believe that the hematite and magnetite additions showed a similar pattern to the natural controls probably as their natural abundance in the sediment promoted the maximum potential of the AOM, as written above.

L379: I think the result of the addition of molybdate is not super surprising. It appears that the molybdate was added rather late in the experiment when the trends are already supporting magnetite as a potential AOM electron acceptor. Sulfate reduction would be naturally inhibited since metal oxides yield much higher free energy than sulfate does in the redox cascade. Yes, we agree. However, molybdate was also added to the treatment without an electron acceptor. If any kind of $SO_4^{2-}$-dependent AOM component existed, the increase in the $\delta^{13}$C-DIC values would have been stopped by molybdate addition.

L385-386: This is interesting but also not super surprising because I believe many sulfate reducers are also iron reducers. Dig into the literature and see if any of the sulfate reducers you detect have been shown to conduct iron reduction. We are aware of this, and and in fact, we are currently working on a project with a sulfate reducer that can reduce iron as well. We believe that at least some of the Fe(III) in the natural sediments are being reduced by sulfate-reducing bacteria. Nevertheless, this does not contradict the fact that at first impression it would be expected that when sulfate reducers are present, they will reduce sulfate. Therefore, we are mentioning their presence in our slurries here while discussing the possibility of $SO_4$-AOM. We added a sentence to the text regarding the potential of the sulfate-reducing bacteria to reduce iron as well.

L392-393: I don't think you can totally confirm that nitrate and nitrite is inhibitory. Fig 3C shows some buildup of the 13C-DIC and again I would be more convinced that there is a true inhibitory effect of the nitrite, if the trends were identical to the killed control. But even then, what evidence is there that nitrate or nitrite inhibits the enzymatic pathway in AOM, like BES does? Does the literature have any suggestions? In addition, you also added hematite to those samples with nitrate in Fig. 3C so how do you know that the buildup of 13C-DIC that you do see is from denitrification coupled to AOM or iron reduction coupled to AOM? Did you measure a buildup of N2 in parallel? Were you able to somehow inhibit iron reduction (if you did that would be cool and would love to know)? As we answered above, we believe that we witnessed the microbes favoring a different process than methane oxidation. Methane oxidation is still occurring in these treatments that show lower enrichments than the treatments without an electron acceptor, only there is also another process that is now more favorable - so either there is less methane oxidation than the "baseline" treatment because of different microbial populations, or there is much more oxidation of organic matter that dilutes the isotopic AOM signal. We understand the confusion over the term "inhibition", and we have clarified it in the text.

We did not measure $N_2$ buildup (the headspace was mostly $N_2$), and it is very challenging to inhibit the iron reduction, at least not the actual iron reduction pathway.

L399: If AQDS has a high electron shuttling capability leading to higher organ clastic turnover then in a closed system like this wouldn't your 13C-DIC become very deplete (Rayleigh distillation) over time and not just plateau like your data suggests? I rather think AQDS just doesn't support anything since it looks just like your killed control or else it would have looked like some kinetic process if any biology was involved. The addition of AQDS did result in a slight depletion of the $\delta^{13}C_{DIC}$ values (it is not visual because of the scaling). It's a depletion of about 17 ‰, and we see an increase of about 70 µM in the iron concentrations while without AQDS there is an increase of only 30 µM (not presented). This suggests that the AQDS supports organoclastic iron reduction. This is now discussed in the text, and a graph of the Fe(II) concentrations was added (to the supplementary information).

L 406: I really think your Fe(II) data belongs in the main text especially here where Figure 3F really needs S3 to support your claim. We accept the referee's suggestion and have moved figure S3 to the main text.

L412: I think it would be worthwhile to have spent a bit more time on magnetite as another potential electron acceptor since Figure 3A is convincing enough, though the scaling in the y-axis is deceiving. We discussed this potential further in the revised version.

L446-448: This was left open ended. What does trace methane oxidation have to do with your study? Plus your experiments are in slurries over long periods of time with amendments that are probably much different that the natural environment so would trace methane oxidation be a likely occurring thing in a slurry. In our two-stage experiments, the net methane-related process is methanogenesis. However, we still observe an enrichment in $^{13}$C-DIC when we add $^{13}$C-labeled methane. Because of that, we cannot say that this is standard AOM. Other studies that observed similar methane oxidation when the net is methane production called it "trace methane oxidation" (Moran et al., 2005; 2007).

L488: You mean aerobic methane oxidation is decoupled from iron reduction right? That is what we meant, but we have rephrased it better in the text, which now reads: "It appears that methanotrophic bacteria cannot survive the long-term slurry incubations and thus iron reduction and aerobic methane oxidation are decoupled"

---

## Author Comment (AC4)

Response to referee #4 (our answers in blue):

In this study, the role of AOM in Lake Kinneret sediment incubations was explored. Several incubations tested which terminal electron acceptors accounted for AOM activity. The main findings were that:

- pre-incubation with methane for 3 months significantly increased AOM
- hematite seemed the most likely iron mineral used as terminal electron acceptor (TEA) for AOM although it did not stimulate AOM and other iron minerals could have inhibited AOM
- natural humic acids and black coffee could be TEA for AOM
- sulfate-AOM was determined neglectable
- BES inhibition indicated that archaea mediated AOM, which was supported by metagenomic and 13C-lipid analyses

Major comments

It will improve the study to have its goals clarified in the introduction by L84-95. What were the specific research questions and knowledge gaps addressed here? What is this study addressing that was not known from your previous studies? This was still not clear to me after reading the entire manuscript. I think this is reflected in the title: "Modification of methane oxidation pathways during long-term incubations of methanic lake sediments" - I could not understand which modification occurred (bacterial methanotrophs did not thrive? TEA changed?). Think about a more specific title that summarizes the main key message of the study.

We thank the reviewer for the thorough and constructive review. As mentioned above, we realize that the paper was not clear enough. In the revised version, we clarified in the introduction the gap of knowledge, the aims and the hypothesis. We also edited the paper throughout. We also accept the comment about the title and changed it to: "Long-term incubations provide insights into the mechanism of AOM in methanogenic Lake Kinneret sediments." This title summarizes the changes from oxygen and iron-oxides being the electron acceptors for methane oxidation, used by bacterial methanotrophs and methanogens, to potentially iron oxides and humic substances used by methanogens.

The materials and methods section needs major improvements for experiment reproducibility - adding amounts, concentrations, units, calculations etc. Sequencing data must be made available and an accession number must be provided. Data that has been already published and is here reproduced must be made clear.

In the revised version, we supply more details on the methods, so every treatment could be reproducible and clear. We accepted all comments below and edited the methods section completely. We moved Table S3 containing the details of the experiments to the main text, and a protocol for each experiment was added to the supplementary material.

All these incubations were done but no methane oxidation rates are provided in the manuscript, so calculating them and presenting them would add a lot of value.

We accept and appreciate this comment. We have added a new table with the different rates of AOM in all the experiments where methane concentrations were measured.

Metagenomics results were barely used (same goes for lipid data). Consider doing metabolic reconstruction of the MAGs recovered here or use this data for another study that explores metabolic potential and mechanisms of potential taxa responsible for Fe-AOM.

We now present in the main text the microbial data and discuss it more thoroughly. We added a principal component analysis biplot to the main text to discuss the difference between the three types of experiments and indicate the dominant taxa. Based on comments of another referee we decided not to explore the metabolic potential based on MAGs here but will do that in a separate publication.

Detailed comments

Review grammar of the manuscript. The manuscript has been edited by a native English speaker collaborator.

L42 - ANME between parentheses. Parentheses were added.

Intro: add background on the black coffee experiment - what was the hypothesis and the literature background? The black coffee was coffee grinds that were added as an organic source, however this treatment was removed from the manuscript, as suggested by the other reviewers.

2.1 add geographical coordinates of sampling site Coordinates were added.

2.2 indicate that concentrations of substrates in pre-incubated sediment experiments are provided in table S1 but bring this table to main text given that it is vital for the manuscript and experimental reproducibility. Also, add to this table similar details about the other two types of experiments (semi-bioreactor and incubation with recently collected material) which are so far missing from the methods section. Indicate if substrates were bought or synthesized (especially for minerals) with manufacturers / synthesis protocols. The methods section was re-organized as was similarly suggested by the other referees; the detailed table (Table S2) was moved to the main text. Details about the freshly collected sediments experiment and the semi-bioreactor were added to that table. The substrate data was added.

2.3 Name it "Porewater and gas analyses"? The name was changed altogether to "analytical methods".

L202 can you provide methane detection limit in total amount (µmol) instead of concentration (µM)? Also, add the volume of gas injected into the GC? The detection limit was provided in total amount and the volume injected to the GC was added.

Eq 1 and 2: provide units for each term, label eq (1) in L205 and (2) in L206; invert eq (1) so it will be ð• • ¹ð• • ·ð• • ¼13ð□□¶ð□‴ = ð□'¥ × ð• • ¹13ð• • ¶ð• • »4 + (1 − ð• ' ¥) × ð• • ¹ð• • ·ð• • ¼13ð• • ¶ð• ' – Units were added. The equations labels were added. There seems to be a problem with the referee's text, we could not read the suggested equation.

Also, can you add what was the final time used to calculate rates? Were rates derived from the slope or from the difference between T0 and T-final? Yes, all details appear now in the text and the protocols.

L161, section 2.2.2 - add bioreactor volume and manufacturer information? The bioreactor's volume was 0.5 L and the bioreactor's manufacturer is LENZ, Weinheim, Germany with custom-made lids. This detail was added to the text in the methods section of the semi-bioreactor section.

2.4 at L215 needs more details for experimental reproducibility: what was the sample exactly

(sediment? how many g?), concentrations of added compounds and steps - protocol format given that a modification of Sturt et al., 2004 was used. Suggestion: release the step-by-step protocol as supplemental material or zenodo link with doi number. Deposit sequencing data and add a data availability statement. Thanks for pointing this out. We added information on the amount and type of sample chosen for lipid extraction as well as the number of internal standards used. The text now reads: "A total lipid extract (TLE) was obtained from 0.4 to 1.6 g of the freeze-dried sediment or incubated sediment slurry using a modified Bligh and Dyer protocol (Sturt et al., 2004). Before extraction, 1 μg of 1,2-diheneicosanoyl-*sn*-glycero-3-phosphocholine and 2-methyloctadecanoic acid were added as internal standards." The extraction protocol in Sturt et al. is a modification of the original Bligh and Dyer method (1959), which itself is a modification of the preceding Folch et al. method (1957).

2.5 How were counts per million reads calculated? Add formula to methods here. Also, can you briefly list all tools that produced data part of this manuscript and are part of the SqueezeMeta pipeline? For instance, what did you use for MAG taxonomic classification? And for genome annotation / gene search? We now use coverage values instead of TPM, as suggested by the reviewer and used GTDB to classify MAGs. This is now clarified in the text as follows: "GTDB-Tk was used to classify the metagenome-assembled genomes (MAGs), based on Genome Taxonomy Database release 95 (Parks et al. 2021)". Following suggestions elsewhere, we now refrain from any analyses involving annotations/gene searches and will discuss them in a separate publication.

I could not fully understand if and which results presented in this manuscript are already published (i.e. L115-117, L249-253). Can you please clarify this? Also, given that a number of different incubations were performed, I suggest numbering them consistently in text, tables and figures to facilitate tracking. The results that were published are the batch incubation experiment with the freshly collected sediments (the batch long term and the fresh bioreactor are new). This was clarified in the text. The different incubation experiments have been numbered.

L265-273 & Figure 2 = the most useful to me would be a plot of methane oxidation rates as a figure and, in the text, something like this: "treatment X or addition of X increased methane oxidation rates (in nmol/dry g sed/day to allow comparisons with other studies/settings) by X% relative to controls". Also, in Fig 2, what is the difference between blue, red and yellow? Add this information in the legend. We added this information to the legend. Following the comment we also added a table where the rates from each experiments are presented.

Fig 2, 3 and 4 = Is it possible to improve the quality? Also, it would be great to have methane oxidation rates in the text or as a figure - from all these different incubations, the only number provided is "3-8 % of the 13C-methane" in L454, which should be presented as a rate - this information I would find most valuable from this study and would allow comparisons with data from other environments, which could be added to the discussion. The quality of the figures is low because they are embedded in the word file. The quality of the original figures is much better, and we will upload them as figure files. The data "3-8% of methanogenesis" is provided to show that even though the net process is methanogenesis, there is still substantial methane oxidation in the slurries. We wanted to compare to other studies that showed methane oxidation in a net methanogenesis environment. However, we agree with the referee, and we added a table with the rates of methane oxidation.

3.2 I suggest showing metagenomic results in the main manuscript. My suggestion is to make a heat map with MAG coverage normalized by metagenome size (instead of RPKM values) and add to this figure the info of Table S3. Also, instead of binscore, use MAG completeness and contamination (in %). Would also be good to know how many MAGs were reconstructed and which ones represent candidate iron reducers - FeGenie could be useful for that: https://doi.org/10.3389/fmicb.2020.00037. We now show a principal component

analysis biplot that shows changes in beta diversity and indicates the dominant taxa (Figure 5 in the main text), and describe it in much more details. Based on comments elsewhere, we decided not to include the metabolic potential based on MAGs, and we discuss it in a separate publication.

Table S4 I was surprised that mcrA and pmoA are not in this table! I think including these and iron reduction and extracellular electron transfer genes would be better use of your metagenomic datasets, which could be extensively better explored in this study. The updated table includes taxonomy only of much more MAGs (~195). As mentioned previously, we will explore the metabolism in a separate manuscript.

L328-331 The numbers here do not match Table 1, which shows more data than discussed here. Maybe this table is not so important and could go to supplemental materials? The lipid biomarker data are important for the study. To clarify the misunderstanding of the reviewer between relative $^{13}$C-enrichment and absolute $\delta^{13}$C values given in Table 1, we rewrote the text that now combines the two lines of information.

Table 1 = Can you clarify what exactly each incubation is and what are killed controls potentially present here? We clarified which incubations are presented in Table 1. Killed controls were tested for lipids in our previous study (Bar-Or et al., 2017) and showed indeed no enrichment. They are not presented in this table.

MAG coverages indicate Bathyarchaeota could be mediating Fe-AOM or play an indirect important role given that they are more abundant than ANME-1 - here the metabolic reconstruction of these MAGs would be fundamental! No mcrA was found in Bathyarchaeota - did you use an HMM that could find divergent sequences? what about other genes in reverse methanogenesis? what is Bathyarchaeota's metabolic potential in your incubations? We agree! As proper analyses of MAGs are expected to inflate this paper drastically, we are keeping this discussion for the follow-up study. We indeed intend to use HMM profiles.

From table S1 I assume hematite is the dominant iron mineral in lake sediments, is it? Then I find curious that this most promising terminal electron acceptor did not stimulate Fe-AOM while other iron minerals could have even inhibited AOM. Can these results alone be taken as evidence for Fe-AOM? I find them insufficient. More discussion is needed to hypothesize about what is happening and how to improve experimental conditions. The Fe-AOM was suggested in our previous studies with fresh methanogenic sediments of Lake Kinneret (Sivan et al., 2011; Bar-Or et al., 2017). In our current study, we examine if and how the methane oxidation changed in the two-stage incubated sediments (meaning that by the time the experiments were set up, the sediments are no longer fresh). In the two-stage incubation, we do not see a difference in the $\delta^{13}C_{DIC}$ between treatments with or without hematite, suggesting that either there is enough hematite to sustain the Fe-AOM, or that it is not the electron acceptor used for the AOM in these slurries. We agree that these results are insufficient to say that this is Fe-AOM, however, we cannot rule this option out. We elaborated the discussion regarding what is happening and added to the text ways to improve the experimental conditions.

In the semi-bioreactor experiment, why was little methane provided (when the methane headspace was replaced by anoxic liquid)? For how long were these semi-bioreactors operated? ~600 days? Also, any particular reason for calling them "semi" and not simply "bioreactors"? Finally, know that from our experience shaking biomass/sediments disrupts

AOM activity (related to L166-7). So, shaken and with little methane, I am not surprised to see in Fig 2 that there was no AOM detected in the bioreactor. In this manuscript, there is no discussion of bioreactor results, so I suggest to add something. We called it "semi" because it represents "semi-continuous flow" as porewater was exchanged weekly to biweekly during sampling. The initial dissolved methane concentration was established by temporarily creating a headspace of $^{13}CH_4$. After 24h equilibration time the headspace was replaced by anoxic porewater. This was the only way to add labeled methane to the reactor. The reactors operated for 677 days. We shook the system at the beginning when the methane head space was created, and after that only before sampling to make sure that the concentrations of the different constituents are homogenous. We did the same before sampling the bottles of our batch experiments and we never experienced any problem. We added more discussion regarding the bioreactor results.

L423 To enrich the discussion on 13C assimilation into lipid, I suggest addressing your results in the context of these findings and potentially more:

Wegener G, Niemann H, Elvert M et al. . Assimilation of methane and inorganic carbon by microbial communities mediating the anaerobic oxidation of methane. Environ Microbiol. 2008;10:2287–98.

Kellermann MY, Wegener G, Elvert M et al. . Autotrophy as a predominant mode of carbon fixation in anaerobic methane-oxidizing microbial communities. Proc Natl Acad Sci. 2012;109:19321–6.

Julia M Kurth, Nadine T Smit, Stefanie Berger, Stefan Schouten, Mike S M Jetten, Cornelia U Welte, Anaerobic methanotrophic archaea of the ANME-2d clade feature lipid composition that differs from other ANME archaea, FEMS Microbiology Ecology, Volume 95, Issue 7, July 2019, fiz082.

We thank the reviewer for these suggestions to support the fact of DIC assimilation by ANMEs/methanogens in our case. We have added these to the manuscript.

L426 move to results It was moved.

L426 Just because ANME are not very abundant it does not mean they are not (very) active. Here abundance is expressed as "< 1.5 %" - specify what this number refers to (relative abundance? how was this calculated? add to methods). We now report the abundance as coverage (~1 for ANME in all the metagenomic libraries). Their abundance is indeed low, but substantial (in top 27 of 195 MAGs, now presented in the Results). We certainly consider them as performing the AOM, and we suggest it in the text. For example, in the discussion we state that "ANME-1 are the likely mediators of AOM in these sediments, although some methanogens may be capable of oxidizing methane too through reverse methanogenesis (Elul et al. 2021)."

L443 I think it's appropriate to tune this down: "we hypothesize Methanothrix could be involved in Fe-AOM". High potential when ANME-1 is present and other archaea are more abundant is a bit stretching; but it would be nice to see some actual physiological evidence for the involvement of Methanothrix in Fe-AOM in the future. Here your back flux inferences also support ANME-1's role being much larger than Methanothrix. We removed the statement regarding Methanothrix, and as mentioned above, emphasize the potential

involvement of ANME-1.

L469 Table S6 is for the first time mentioned here in the discussion. It presents qPCR results that have not been mentioned in the methods, so these must be added and the mention must be moved to results. Methanogenesis rates are expressed in µM/day, which I found cryptic and does not allow comparisons to other studies - please convert to n or µmol/dry g sed/day. The qPCR results of the mcrA gene were taken from the Bar-Or et al., 2017 study in order to provide a general order of magnitude estimation. The rates were calculated in this study. This was clarified in the text. The rates of methanogenesis were converted as suggested.

L470 I am missing and thus suggest adding a sentence hypothesizing about the key microorganisms (ANME-1) accounting for 3-8 % of 13C-methane oxidation to CO2 in these incubations. Also, what is this number referring to? Hematite-AOM? Humic acid-AOM? I would love to see rate comparisons between those! The sentence was added to the text as suggested. The number is referring to the slurries without any electron acceptor addition. We wanted to calculate how much of the produced methane is being oxidized in the basic environment of the slurries. However, as the referee suggested, we added the rates of oxidation in other treatments as well.

L481-8 I find this insufficient to explain why putative bacterial methanotrophs disappeared in long-term incubations if oxygen could be generated via methanobactins. However, this must be stated at hypothesis level, we don't know if iron reduction and methane oxidation were coupled via methanobactin-produced oxygen. I think it's better to offer other explanations or simply say it's unclear why bacterial methanotrophs disappeared. We do not know if $O_2$ is in fact produced in the natural sediments via methanobactins or another method, however, the fact that we do not see enrichment in their biomass in our two-stage experiments suggests that aerobic methanotrophy is no longer occurring in those slurries. The metagenomic analysis shows that their copy numbers are very low and do not increase with time. This implies that they are not active. Indeed, we do not know if the reason for that is lack of $O_2$, even though that is the most reasonable answer. This was clarified in the text.

---

## Referee Report (RR1)

Vigderovich et al., in their revised manuscript titled "Long-term incubations provide insight into the mechanisms of anaerobic oxidation of methane in methanogenic lake sediments" made considerable changes to the previous version. I believe the results in the manuscript are still interesting and appropriate for the Biogeosciences journal, however, I still think there is major work to be done before publication.

The manuscript needs attention in its consistency, flow, redundancy and clarity. The scientific English has gotten much better since the last version, however, there are still some syntax and sentence structure issues that make it difficult for the reader to follow. The manuscript also taps into results and interpretations from previous work from the research group and needs to do a better job of clarifying which interpretations are from THIS manuscript vs. previous work (i.e. fresh batch incubations from Bar-Or et al.,). I again cannot identify them all, but I will provide examples below. I highly encourage the authors to consider using an editorial service to address the English in the manuscript.

Major Comments:

The introduction provides known facts about electron acceptors known to be coupled to AOM. However, it lacks general information about AOM occurring in lakes and mainly compares the marine environment generally. I suggest adding a few sentences about AOM in lakes or freshwater systems (i.e. lower sulfate concentrations, more methanogenesis, larger role of metals and nitrogen etc...) generally would help with flow and make the introduction more robust. The manuscript also lumps ANME and methanotrophic bacteria but does not introduce aerobic oxidation of methane and bacterial facilitators very well. I suggest making two separate paragraphs to describe anaerobic and aerobic oxidation of methane. The introduction now has a statement telling the audience the knowledge gap; however, I feel that the statement could be crafted better by being more specific about which methanotrophs and what kind of sediments (i.e lake sediments). Furthermore, the last paragraph of the introduction is already in the methods section and should either be taken out or greatly shortened.

The most challenging part of the manuscript is the methods section. The experiments are difficult to reproduce based on what is written here. I acknowledge the protocols found in the supplementary are useful, but the information found in the main text takes precedent. For example, there is hardly any information about how many cores were taken and when exactly they were taken between the years 2017 and 2019. It is also not clear when sediments for experiments type B and C were taken; there is one sentence that implies sediments were collected in 2013 which is confusing. Additionally, the text very sparingly cites the supplemental to direct the reader for more information about the protocols. If the authors wish to have the reader go to the supplemental for this information rather than put it in the main text it should be cited accordingly and in order.

The results now do a much better job of simply stating the data without interpretation than the previous version. However, there are several syntax and other issues which I identified some examples below in the in-line comments.

The discussion would benefit greatly with proper headers for all sections. It would also benefit with references that backup the interpretations presented, please see below for more details. The discussion would further benefit by adding sentences that conclude their interpretations for each section. There are parts of the discussion that beautifully build up what was done to help interpret the meaning but then ends rather abruptly with no resolution or even speculation of what maybe happening (see below for more details).

In line comments and edits:

L38-41: What about AOM in lake sediments?

L42: This first sentence is rather redundant considering the last sentence of the previous paragraph. Consider joining the two paragraphs to be more concise.

L45: What other metals are you referring too?

L52-55: I think 1 or 2 more sentences introducing aerobic oxidation of methane and aerobic methanotrophs should be included here. What is their difference to their anaerobic counter parts? The paper is about how the two processes are potentially co-occurring which is unusual so it would be useful to introduce the typical conditions these processes occur in lake sediments such that the reader understands the "normal" and what is not normal.

L42-55: This paragraph could have better flow. The authors just state facts about different electron acceptors but do not connect how these electron acceptors are coupled to AOM in lake sediments. For example, L51: what does selenite reduction have to do with AOM in lake sediments? How is this a supporting fact?

L50: Syntax error: "whereas nitrite fuels AOM by Methylomirabilis" does not make sense.

L56: There needs to be a better introductory sentence for this paragraph. Or better a concluding sentence(s) in the previous paragraph that connects AOM in lakes and then have a good intro sentence about Lake Kinneret.

L60: Are you sure the archaea are methanogenic or are they methanotrophic, not clear in the sentence?

L63: First time that type I aerobic methanotrophic bacteria are mentioned. Please see comment above about providing more introduction to aerobic oxidation of methane.

L63: What role are the playing methane cycling? I assume they are working together to oxidize methane in the lake sediments, not clear.

L64: Syntax error; how do you have aerobic methanotrophic activity in anoxic environments? This statement implies an aerobic process occurring in anoxic conditions which is contradictory. Do you mean "The combined methanotrophic activity by anaerobic archaea and aerobic bacteria in anoxic lake sediments might be supported by the presence of microlevel oxygen…."?

L66: Please provide a number for microlevel. Also there needs to be references that support the second half of the statement.

L67-68: Which methanotrophs do you mean here? Do you mean both the ANME and bacteria? I assume this trend is not well understood in lake sediments either? I see this is your knowledge gap statement and this statement could be stronger by being more specific.

L69-84: Most of this paragraph is redundant and already in the methods section. I understand the purpose of the paragraph, but it is far too long and should be brief.

L87: "in the North of Israel" implies the lake is not in Israel. Northern Israel would be more correct.

L90: From April till when?

L90-91: Are the surface water between 15-30 degrees C all year long? Or are certain months 15 or 30?

L91-94: What sediment depths is this data referring too? This sentence also seems redundant since in L94-96 reports nearly the same data in table S1.

L92: Are the references for clays in this sentence only for clays or both clays and carbonates? If the ladder where is the reference for the carbonates?

L99: At what sediment depth does it get as low as 0.5 mM?

L104: Inhibitors for methanogenesis not methanogens.

L105: Is it necessary to say this is a "Long-term" incubation? All of the incubations seem to be long-term, why not just say two stage?

L105-106: Sentence is hard to follow, please restructure.

L107: The slurry was further diluted with what?

L109: How fresh is freshly? This is said a lot throughout the manuscript, but it is not clear what fresh means. Where are these sediments from? What sediment depth(s) are these sediments from?

L112: If you say several manipulations in this fashion then I would expect a description of each manipulation because this is rather vague. But since they are manipulations from previous work just say "and amended with… according to (REFERENCES)…"

L117-122: I think this paragraph needs to be its own subsection to describe how sediments were collected for all your experiments. Right now I have to assume this is how sediments were collected only for the "Long term two stage incubation". Then the question is how were sediments collected for types B and C?

L117: Could you please specify how many sampling campaigns there were between 2017 and 2019.

L118: Was there a research vessel involved, if so which one?

L118: Syntax error: how does one use a gravity corer with 50 cm Perspex cores? Do you mean equipped with 50 cm Perspex core liners? How many cores were taken? What was the total length of sediment collected? Was it the same amount taken every time? How long did sediments sit in the core liners before processing?

L120: How many parallel cores?

L120-122: Was sediment sliced into intervals and put into 50 mL conical vials for centrifugation and then pooled later? There are some details missing as to how the sediments were processed for porewater extraction.

L121: please add a times symbol between 9300 and "g".

L123: How much sediment? How was the sediment sampled from the core liner (i.e. slicing, cut-off syringe, bulk transfer)? Was the sediment added to the porewater or was porewater added after sediment was sampled? Were these also sealed with stoppers and crimped?

L125: Please provide product details for methane that was injected; similar to the nitrogen.

L123: I think you already said this in L119. I am confused, how many slurries are there?

L127: I think there is a syntax error. Is there a reason to have two experimental pathways when significant AOM is observed? Or do you mean that if no significant AOM is observed in the slurries then porewater is exchanged and continued to be monitored? Not clear.

L130-138: This information could be better integrated in the beginning of this section.

L130: I am still confused about the sampling in this section. There was 2 years that sediments were collected, and 10 sets were made for this experiment. When was each set collected? Did each set run in parallel with each other or did they run independently and different times?

L132: Is the pre-incubation the first stage? Please be consistent with your identifiers.

L132: Was the 1st stage bottle opened and then sampled? Not clear.

L137-138: Did your killed controls have any sediment in them? This sentence reads as if only the bottles were autoclaved and 13C-methane was added to them.

L142-144: Id rather like to see how AQDS was prepared for your experiments and not what it has been shown to do; which was already introduced in the intro.

L139-154: This paragraph would read better if you included which experiment number or set corresponds to which electron acceptor or shuttler like you do with the inhibitors in the end of the paragraph.

L151-154: Are experiment numbers the same as "sets" in L130? Please be consistent with your identifiers.

L151: Is there a reason why molybdate was only added to the magnetite samples?

L155: A sentence after this one describing why duplicate or triplicates were made would be useful.

L159: Please provide which lake and why did you pick a different lake to get humic substances.

L161: So far I have not seen how your geochemistry (iron (II) and methane) are measured. I assume there is a section for that but perhaps add a parenthetical to indicate that.

L167-168: When were these sediments collected and by what method. Were they collected by gravity corers like experiment A? Please elaborate. If all sediments for all experiments were collected between 2017 and 2019 by gravity corers etc… then that should be stated before all the experiments are described.

L169: Similarly, with the porewater. Is this porewater from the same methanogenic depth? Please add details or blanket the information before all the experiments are described.

L171: What is the relative proportion of the 12C and 13C in the mixture?

L174: What batch experiments?

L175: Using an electrode not by electrode.

L182-183: Are you saying the pre-incubation sediments were collected in 2013 instead of between 2017 and 2019? The timing of sample collection for the sediment samples for any of these experiments is not clear at all. I suggest that there should be a new section in the beginning of the methods describing when sampling occurred for which experimentation.

L185: Please provide the material of the experimental bottles.

L190-195: There is information in the figure caption that should be in the main part of the methods. Please move them accordingly.

L198: I suggest this section needs to have subsections to disambiguate between geochemistry and molecular analysis. It is confusing to read about geochemical analysis and then jump to molecular analysis and then back to stable isotope analysis. Having sub headers would help the reader not only know when you are switching gears but also, for example, if someone wanted to find quickly a detail about the molecular analysis without reading the whole manuscript, they could easily find it using the sub headers.

L206: Determined not "measured". You measured on a spectrophotometer.

L207-210: These two sentences do not read well. One could easily put them together for better flow.

L208-210: Please provide the GC model, column type and carrier gas.

L210-211: Ethylene was similarly determined to what? You mean that ethylene from acetylene reduction was measured by GC right?

L231: Not clear what is meant by duplicates a and b in the semi-bioreactor experiment.

L235: Never good to start a sentence with a numerical. Consider starting with "A range of …" or spelling out nineteen.

L245: Rate should be plural. Please check tenses throughout.

L262-263: Syntax error. This sentence reads as if the results from type A and B were compared to the batch slurry incubation presented by Bar-Or et al., and Elul et al.,. You mean that you are comparing YOUR type C experiments with Bar-Or et al., and Elul et al., type C experiments, correct?

L266: Which samples do you mean when you say natural non-killed slurries? Figure 2 and 3 do not have any identifiers called "Natural non-killed slurries". Do you mean all the slurries with different amendments? If so then I would not consider them natural. This is not clear and very confusing.

L266: How "significant" is the AOM and relative to what? Did you do any statistics? I would be careful using this word.

L266-268: Sentence does not read well, please restructure.

L270: I think that it is interesting that you have such a methanogenesis rate and Figure S3 should be moved into the main text. Also Figure S3 reports a rate in µM in the figure caption, mM in PW on the S3 y-axis and in nmol gr^-1 Day^-1. I suggest making it all one unit type as it is very confusing.

L271: Now it is a two-stage incubation and not long term please be consistent.

L275: What treatments? Do you mean replicates? Figure 2 suggests that they are replicates.

L274-276: Sentence could be structured better.

L282: Please move the Figure 3B citation to just after the describing the increase from the nontrite.

L286: AOM is not "of" the incubation it is "in"

L287: Was sulfate measured in this study? I do not recall it in the methods.

L289: I do not think that molybdate inhibits the sulfate reducing bacteria, it rather inhibits sulfate reduction by acting as an analog to sulfate and bind to APS enzyme. Please be careful with distinguishing organism with metabolism. This has also already been stated elsewhere, consider removing.

L290-291: This sentence does not read well. Please restructure.

L290-291: This is not so surprising since magnetite was added and iron reduction is not inhibited by molybdate and it outcompetes with sulfate reduction. If there is no sulfate/sulfide concentrations reported nor any rates, then does this have a place in this manuscript?

L294-297: First time nitrate/nitrite measurements were mentioned. How did you measure it? This should rather be in the methods.

L301-303: This sentence does not read well please reorganize.

L303-304: Is this AOM rate for nitrate or nitrite coupling?

L320: Again BES is not an inhibitor of the methogen archaea it is an inhibitor of methanogenesis. You also need references.

L325: Methanogenesis not methanogens'

L327: What does the (SN-#) in figure 2 mean? I think that part should be taken out since it has no meaning for the reader and could lead to some confusion. Also why arnt the rest of the 10 sets of the long term two stage results in the graph. Isnt the point to compare all three experiments in its entirety?

L345-348: Table 2 is a bit confusing. What is the serial number? How is there multiple treatments per serial number?

L375: Figure 5 is oddly placed after the results section of the molecular analysis. I suggest moving this as to not confuse the reader.

L399: Get rid of "our"

L401-405: I think you should put the respective citation to the respective analysis in this sentence. For example, Figure 2 is not a representative of in-situ geochemical or microbial diversity profiles. Also this statement is rather generalized, are you saying this is happening everywhere or at Lake Kinneret?

L405-407: Are you talking about previous work with profiles or your results, not clear. If it is others please provide references.

L399-411: I am having a tough time distinguishing what is just a rereporting of findings from older publications and that of the interpretations of the present study. The point of the discussion is to interpret the meaning of results of the present study and use previous lit to support the argument.

L416-417: It is weird to cite a different publication to describe your result. Instead of citing your figure along with a different reference, why not use the space to compare the findings in the present study with that of Bar-Or et al.,.

L416-422: Why is the B and C experiment being discussed here when the header suggests that the Long-term two stage incubations will be discussed? It seems like here you are comparing all three experiments rather than focusing on interpreting "Potential electron acceptors for AOM in the Long-term two-stage incubation experiments". This header is misleading as one of your experimental types (A) is called Long-term two-stage incubation.

L419-420: I think that the differences in the bioreactors are interesting, but I also think that it is not super surprising as slurries in replicate can act as independent communities, because they are removed from the original environment, amended and sit for such a long time.

L420-422: I also don't understand this statement. According to your data AOM was stimulated with hematite better than all the other amendments that were done.

L420: Again, did you do any statistics to suggest that it is significant?

L425-428: Any lit to support that explanation?

L428-432: Run-on sentence, hard to follow and confusing please trim and reword.

L436-437: I am not sure your data suggest that adding iron decreases AOM activity directly. It is rather better to say iron is just being used for organoclastic iron reduction rather than to oxidize methane.

L437-441: You need references to back these statements up.

L442-451: See comment to L290-291. I think this should be omitted or greatly reduced and stated in the beginning that you believe sulfur cycling does not play a role in your experiments.

L453-454: Ammonium oxidation with iron reduction needs a citation.

L455-459: I understand your results don't indicate nitrate supports AOM but I don't understand from your interpretations how nitrate delays AOM. Why would organoclastic denitrification outcompete AOM? There is no citation in the text that supports that claim. Denitrification would of course be outcompeting other organisms for organic material but AOM is only oxidizing methane, which you added in abundance. I would agree that it is strange to have no 13C DIC build up after adding nitrate but potentially, the addition of nitrate resulted in more organics to be oxidized a dilute the 13C signal. Furthermore, the text clearly stated ANME-2d was not found. So I think that it is more accurate to say that nitrate does not support AOM as an electron acceptor during your experiments because the known ANME group that uses nitrate was not present. Also figure 3 C and D do not show any trends with sediment amended with nitrite.

L461-463: This sentence is contradicting. You said they weren't directly measured but then they were high. It should rather say previous work has shown DOC concentrations to be high in Lake Kinneret…

L465-467: References

L467: Cite your figure here. This is the one hit you got out of all the experiments and therefore, the crux. Please elaborate and compare to the Valenzuela et al.,

L479-482: Run-on sentence. Reads poorly.

L487-497: Do the authors have any interpretations as to why the aerobic methanotrophs play a minor role? I think this paragraph could be much stronger a it is eluding to who is responsible for oxidizing methane. Consider structuring the paragraph to better nail who is overall responsible.

L520-521: The authors built this paragraph up nicely but if it is is not back flux by methanogens then what could it be? This kind of ended abruptly.

L529-531: This info could be useful in the discussion.

---

## Referee Report (RR2)

In the latest iteration of this manuscript by Vigderovich et al., the authors put considerable effort to the language, flow, and clarity through the whole document. The methods are much easier to follow and are much more reproduceable.  The results are properly reported and discussed. However, I have only one minor comment which I will point to below. Otherwise, the remaining minor comments and edits are rather small and mostly cosmetic (see inline comments).

Minor comment:

The results from the metagenomic and lipid analysis clearly show evidence of aerobic methane oxidizing bacteria. The discussion does clearly state that if aerobic methane oxidizing bacteria do play a role in turning over methane in these experiments is quite low. However, the conclusion reads as if they play a much bigger role in the turnover with methane which is hugely speculative. The problem I have with this portion of the manuscript is that aerobic methane oxidizing bacteria were not directly tested in any of the experimental setups. Oxygen concentrations were not determined, and the experiments were all set up anaerobically. Thus, the incubations were not setup to directly test for aerobic oxidation of methane activity. I still think though the metagenomic and lipid findings are a real bonus dataset and are very interesting and should be investigated further. However, I do not think the results presented in this paper warrant the statement in the conclusion that aerobic methane oxidizing bacteria play a role since it wasn't directly tested. I therefore, suggest that the paper would be much stronger by really highlighting the coupling of hematite to AOM by anaerobic archaea. Following this conclusion, provide examples of how aerobic oxidation of methane by the bacteria maybe involved in Lake Kinneret sediments, but the results presented here show a need for further direct testing of this potential.

Inline comments:

Overall, the manuscript reads much better. Below are some inline edits and comments I caught while reading. It is likely I did not catch all and I suggest the authors go back and thoroughly fix minor grammatical errors.

L54-55: There are other types of aerobic methane oxidizing bacteria. This sentence reads as if there is only 1 in existence. I suggest just adding in parentheticals type II and type X.

L52: get rid of "Thus"

L79: Are you saying Fe-AOM is oxidizing methane in the methanogenic zone or Fe-AOM removes methane produced from methanogenic zone in the top 20 cm?

L88: Include "conditions" after "hypoxia"

L94: Delete "are available"

L100: Add "of " before "two stages"

L127: Please add how much 13C-methane you added.

L137-139: This would probably read better as two sentences.

L150: Cores were sliced at what cm intervals?

L151: Please provide details on the dedicated container. Also why not directly into falcon tubes?

L149-154: I am a bit confused here. You collected cores and extracted porewater on the same day. However, porewater was extracted in the lab while core collection and slicing happened on board. Were these just day long excursions and you were able to bring the sediments back to the lab for extraction? If so I suggest making this more clear that field sampling and laboratory processing happened on the same day of collection.

L153: Syringe filtered or filter tower?

L155: Missing temperature units.

L155: Subsampled not subsamples

L162: Was the N2 atmosphere maintained in a glove bag or continuous flushing? Please add.

L163: Please add how you homogenized the sediment.

L163: Add the word "was" between "gr" and "transferred"

L258: Add spectrophotometer model

L263-264: For ethylene determinations did you use the same column and carrier gas for these measurements? If so consider consolidating into the previous sentence with methane.

L326: How "significant" did you do statistics. I would consider removing.

L350-352: Sentence does not read well.

L352-354: The addition of what had no increasing effect to the 13C DIC pool? Molybdate or magnetite?

L479: How significant of a finding is this. Where are the statistics behind this claim? I would consider removing the word "significant" in the document when statistics was not done.

L485: I think the flow of the manuscript would be better to have subsections to 4.2 for each of the electron acceptors (i.e. 4.2.1 various metal oxides as electron acceptors etc…).

L603: You mention in the discussion that Aerobic methane oxidizing bacteria play a minor role in turning over methane. This was only drawn from lipid and fatty acid determinations and genomic evidence which doesn't necessarily mean they were active. Nor were any of these results connected to the 13C-DIC increases seen in the experiments. I therefore, suggest to adjust this sentence to say that the exact role of aerobic methane oxidizing bacteria in Lake Kinneret sediments needs further examination. Because as is, the text reads that they are more involved than what is reported in the data.

---

## Author Response (AR2)

Response to the reviewer (in blue):

Vigderovich et al., in their revised manuscript titled "Long-term incubations provide insight into the mechanisms of anaerobic oxidation of methane in methanogenic lake sediments" made considerable changes to the previous version. I believe the results in the manuscript are still interesting and appropriate for the Biogeosciences journal, however, I still think there is major work to be done before publication.

The manuscript needs attention in its consistency, flow, redundancy and clarity. The scientific English has gotten much better since the last version, however, there are still some syntax and sentence structure issues that make it difficult for the reader to follow. The manuscript also taps into results and interpretations from previous work from the research group and needs to do a better job of clarifying which interpretations are from THIS manuscript vs. previous work (i.e. fresh batch incubations from Bar-Or et al.,). I again cannot identify them all, but I will provide examples below. I highly encourage the authors to consider using an editorial service to address the English in the manuscript.

We would like to thank the reviewer for the positive, constructive, thorough and thoughtful review. We addressed all of the comments and sent the manuscript to be professionally edited for English. We believe that the manuscript is now ready for publication.

Major Comments:

The introduction provides known facts about electron acceptors known to be coupled to AOM. However, it lacks general information about AOM occurring in lakes and mainly compares the marine environment generally. I suggest adding a few sentences about AOM in lakes or freshwater systems (i.e. lower sulfate concentrations, more methanogenesis, larger role of metals and nitrogen etc…) generally would help with flow and make the introduction more robust. The manuscript also lumps ANME and methanotrophic bacteria but does not introduce aerobic oxidation of methane and bacterial facilitators very well. I suggest making two separate paragraphs to describe anaerobic and aerobic oxidation of methane. The introduction now has a statement telling the audience the knowledge gap; however, I feel that the statement could be crafted better by being more specific about which methanotrophs and what kind of sediments (i.e lake sediments). Furthermore, the last paragraph of the introduction is already in the methods section and should either be taken out or greatly shortened.

All of the reviewer's comments about the introduction were addressed: We restructured the introduction, we added the general requested information (about aerobic and anaerobic methanotrophy, about AOM in freshwater compared to the marine environment and about the involvement of aerobic bacteria in AOM), and we focused our statement about the knowledge gap. We also shortened the last paragraph.

The most challenging part of the manuscript is the methods section. The experiments are difficult to reproduce based on what is written here. I acknowledge the protocols found in the supplementary are useful, but the information found in the main text takes precedent. For example, there is hardly any information about how many cores were taken and when exactly they were taken between the years 2017 and 2019. It is also not clear when sediments for experiments type B and C were taken; there is one sentence that implies sediments were collected in 2013 which is confusing. Additionally, the text very sparingly

cites the supplemental to direct the reader for more information about the protocols. If the authors wish to have the reader go to the supplemental for this information rather than put it in the main text it should be cited accordingly and in order.

All of the reviewer's comments about the methods were addressed. The main text of the revised version of the article contains more detailed explanations about the experiments, and more references to the supplementary material were added throughout to aid the reader.

The results now do a much better job of simply stating the data without interpretation than the previous version. However, there are several syntax and other issues which I identified some examples below in the in-line comments. The discussion would benefit greatly with proper headers for all sections. It would also benefit with references that backup the interpretations presented, please see below for more details. The discussion would further benefit by adding sentences that conclude their interpretations for each section. There are parts of the discussion that beautifully build up what was done to help interpret the meaning but then ends rather abruptly with no resolution or even speculation of what maybe happening (see below for more details).

We addressed all of the relevant comments as detailed below.

In line comments and edits:

L38-41: What about AOM in lake sediments?

We restructured the introduction, and we added organized information about the phenomena of aerobic and anaerobic methanotrophy in lakes compared to marine environments.

L42: This first sentence is rather redundant considering the last sentence of the previous paragraph. Consider joining the two paragraphs to be more concise.

We revised the text.

L45: What other metals are you referring too?

We are referring to metals such as chromium, and that detail has been added to the text.

L52-55: I think 1 or 2 more sentences introducing aerobic oxidation of methane and aerobic methanotrophs should be included here. What is their difference to their anaerobic counter parts? The paper is about how the two processes are potentially co-occurring which is unusual so it would be useful to introduce the typical conditions these processes occur in lake sediments such that the reader understands the "normal" and what is not normal.

We agree with the reviewer and added two paragraphs to explain aerobic and anaerobic methanotrophy processes in different environments.

L42-55: This paragraph could have better flow. The authors just state facts about different electron acceptors but do not connect how these electron acceptors are coupled to AOM in lake sediments. For example, L51: what does selenite reduction have to do with AOM in lake sediments? How is this a supporting fact?

We improved the paragraph's flow by specifically describing each process vis-à-vis the different environments.

L50: Syntax error: "whereas nitrite fuels AOM by Methylomirabilis" does not make sense.

The sentence was changed.

L56: There needs to be a better introductory sentence for this paragraph. Or better a

concluding sentence(s) in the previous paragraph that connects AOM in lakes and then have a good intro sentence about Lake Kinneret.

As suggested, we added an introductory sentence for the paragraph to contextualize our work in the frame of the AOM in general and in the frame of AOM in lakes.

L60: Are you sure the archaea are methanogenic or are they methanotrophic, not clear in the sentence?

Though they are known methanogens, they might perform AOM via reverse methanogenesis. This is now clarified in the revised text.

L63: First time that type I aerobic methanotrophic bacteria are mentioned. Please see comment above about providing more introduction to aerobic oxidation of methane.

They are now introduced earlier as suggested.

L63: What role are the playing methane cycling? I assume they are working together to oxidize methane in the lake sediments, not clear.

The role played by these aerobic bacteria is better explained, with greater detail, in the revised text.

L64: Syntax error; how do you have aerobic methanotrophic activity in anoxic environments? This statement implies an aerobic process occurring in anoxic conditions which is contradictory. Do you mean "The combined methanotrophic activity by anaerobic archaea and aerobic bacteria in anoxic lake sediments might be supported by the presence of microlevel oxygen…."?

Such aerobic activity is observed in many environments that are considered anoxic. It is possible that there may still be trace amounts of undetected oxygen in the sediment or that it is somehow being generated in small amounts. This is better explained in the revised introduction and discussion.

L66: Please provide a number for microlevel. Also there needs to be references that support the second half of the statement.

We added the reference that describes that undetected trace amounts of oxygen (nanomolar levels) may be trapped in "anoxic" environments.

L67-68: Which methanotrophs do you mean here? Do you mean both the ANME and bacteria? I assume this trend is not well understood in lake sediments either? I see this is your knowledge gap statement and this statement could be stronger by being more specific.

We clarified this detail, referring to aerobic methanotrophs in the text.

L69-84: Most of this paragraph is redundant and already in the methods section. I understand the purpose of the paragraph, but it is far too long and should be brief.

We substantially shortened this paragraph in the revised version.

L87: "in the North of Israel" implies the lake is not in Israel. Northern Israel would be more correct.

This detail was changed as suggested.

L90: From April till when?

Until December. This was added to the text.

L90-91: Are the surface water between 15-30 degrees C all year long? Or are certain months 15 or 30?

In the years that were studied, surface water temperatures ranged from 15°C in the winter (January) to 32°C in the summer (August).

L91-94: What sediment depths is this data referring to? This sentence also seems redundant since in L94-96 reports nearly the same data in table S1.

The sentence was removed, and only the previous data that we collected for the methanogenesis zone were cited.

L92: Are the references for clays in this sentence only for clays or both clays and carbonates? If the ladder where is the reference for the carbonates?

The references were for both clays and carbonates, but the sentence was removed.

L99: At what sediment depth does it get as low as 0.5 mM?

At sediment depths 30 cm below the water-sediment interface, the value is as low as 0.5 mM. This detail was added to the text.

L104: Inhibitors for methanogenesis not methanogens.

The term was corrected.

L105: Is it necessary to say this is a "Long-term" incubation? All of the incubations seem to be long-term, why not just say two stage?

Though the reviewer is correct, we wanted to emphasize the lengths of the incubations, which is needed due to the methanogen turnover time of several months.

L105-106: Sentence is hard to follow, please restructure.

The sentence was restructured.

L107: The slurry was further diluted with what?

The slurry was further diluted with porewater extracted from the same depth; this detail was added to the text.

L109: How fresh is freshly? This is said a lot throughout the manuscript, but it is not clear what fresh means. Where are these sediments from? What sediment depth(s) are these sediments from?

Freshly means up to three days from sediment sampling. This was clarified in the text that now reads: "Semi-continuous bioreactor experiments in which sediments were collected up to three days before the experiment was set up (freshly sampled sediments) …". All of the experiments were performed with sediments from the methanogenic zone (depths > 20 cm).

L112: If you say several manipulations in this fashion then I would expect a description of each manipulation because this is rather vague. But since they are manipulations from previous work just say "and amended with… according to (REFERENCES)…"

The sentence was changed as suggested, and it now reads: "Batch incubation experiments with freshly sampled sediments and porewater at a 1:5 ratio, respectively, and amended with hematite. This experimental set-up was described in our previous studies (Bar-Or et al., 2017; Elul et al., 2021)."

L117-122: I think this paragraph needs to be its own subsection to describe how sediments were collected for all your experiments. Right now I have to assume this is how sediments were collected only for the "Long term two stage incubation". Then the question is how were sediments collected for types B and C?

We agree with the reviewer and the text was modified as suggested.

L117: Could you please specify how many sampling campaigns there were between 2017 and 2019.

There were 7 sampling campaigns. This detail was added to the text.

L118: Was there a research vessel involved, if so which one?

Yes, a research vessel, *Lillian*. This detail was added to the text.

L118: Syntax error: how does one use a gravity corer with 50 cm Perspex cores? Do you mean equipped with 50 cm Perspex core liners? How many cores were taken? What was the total length of sediment collected? Was it the same amount taken every time? How long did

sediments sit in the core liners before processing?

The text was corrected as suggested and the missing details were added. The text now reads: "The sediments for the slurries conducted in the current work were collected during seven sampling campaigns aboard the research vessel *Lillian* between 2017 and 2019 from the center of the lake (Station A, Fig. S1) using a gravity corer with a 50-cm Perspex core liner. The length of the sediment in each core was 35-45 cm. During each sampling campaign, 1-2 sediment cores were collected for the incubations and 10 cores were collected for the porewater extraction. Sediments from the methanogenic zone (sediment depths > 20 cm) were diluted with porewater from the methanogenic zone of parallel cores sampled on the same day. The porewater was extracted on the day of sampling. The sediment cores were sliced while onboard, and sediment samples from the methanogenic zone (> 20 cm) were transferred to a dedicated container. In the lab, sediments were collected with 20-ml cutoff syringes and moved to 50-ml falcon tubes. The porewater was extracted by centrifugation at $9300 \times g$ for 15 min at 4℃, filtered by 0.22-µM filters into 250-ml pre-autoclaved glass bottles, crimp-sealed with rubber stoppers, and flushed for 30 min with $N_2$. The extracted porewater was kept under anaerobic conditions at 4 until its use. The sediments for the incubations were subsamples from the liners and diluted no later than three days after their collection from the lake and treated further according to the experimental strategies described above (setup A or B)."

L120: How many parallel cores?

Ten parallel cores. This detail was added to the text. Please see the response to the comment above.

L120-122: Was sediment sliced into intervals and put into 50 mL conical vials for centrifugation and then pooled later? There are some details missing as to how the sediments were processed for porewater extraction.

The sediment cores for the porewater extraction were sliced onboard. Sediment from depths below 20 cm was transferred to a container. In the lab, we transferred the sediment from the container (using cutoff 20-ml syringes) to 50-ml falcon tubes that were then centrifuged. The extracted porewater was kept under anaerobic conditions at 4°C in pre-autoclaved bottles until its use. These details were added to the revised text.

L121: please add a times symbol between 9300 and "g".

The symbol was added.

L123: How much sediment? How was the sediment sampled from the core liner (i.e. slicing, cutoff syringe, bulk transfer)? Was the sediment added to the porewater or was porewater added after sediment was sampled? Were these also sealed with stoppers and crimped?

The relevant data were added to the text, which now reads: "In the first stage (pre-incubation slurry), the sediment core was sliced under a nitrogen atmosphere and sediments from depths > 20 cm were collected into zipper bags. The sediment was homogenized, and between 80-100 gr transferred into 250-ml glass bottles under continuous $N_2$ flushing. The sediments were diluted with the extracted porewater to create a 1:1 sediment to porewater slurry with a headspace of 70-90 ml (Fig. 1). The slurries were sealed with rubber stoppers and crimped caps and were flushed with $N_2$ (99.999%, MAXIMA, Israel) for 30 min."

L125: Please provide product details for methane that was injected; similar to the nitrogen.

The details (99.99%, MAXIMA, Israel) were added.

L123: I think you already said this in L119. I am confused, how many slurries are there?

We clarified this point in the revised text. We added a subsection to explain the sampling procedures at the lake and in the lab, after which we referred to each experiment type for specific dilution details and slurry amounts.

L127: I think there is a syntax error. Is there a reason to have two experimental pathways when significant AOM is observed? Or do you mean that if no significant AOM is observed in the slurries then porewater is exchanged and continued to be monitored? Not clear.

We produced 1:1 slurry to porewater ratio and monitored the pre-incubations. We exchanged the anoxic porewater to supply the natural dissolved nutrients and organics in addition to methane and enrich the community with high sediment to porewater ratio (as in natural sediment, where their abundance is very low). Then we further diluted the samples to enable more microbial enrichment and the extensive geochemical investigation. We also tested the aerobic methanotrophic activity at each stage by collecting samples for the metagenome and lipid analyses.

L130-138: This information could be better integrated in the beginning of this section.

We rearranged the information according to the reviewer's suggestion.

L130: I am still confused about the sampling in this section. There was 2 years that sediments were collected, and 10 sets were made for this experiment. When was each set collected? Did each set run in parallel with each other or did they run independently and different times?

The sediments were collected during the 7 campaigns between the years 2017-2019 (including 2019). Some of the two-stage experiments ran simultaneously and some independently at different times. The details of the dates and the numbers of treatments are clarified in the revised main text and supplementary.

L132: Is the pre-incubation the first stage? Please be consistent with your identifiers.

Yes; we clarified this detail in the text.

L132: Was the 1st stage bottle opened and then sampled? Not clear.

At the end of the first stage (i.e., pre-incubation), the bottles were opened under anaerobic conditions and transferred for the second stage using a syringe and a Tygon tube. These details were added to the text: "…the pre-incubation bottle was opened and subsamples (~18 g each) were transferred with a syringe and a Tygon® tube under a laminar hood and continuous flushing of $N_2$ gas into 60-ml glass bottles. The subsamples were then diluted with fresh anoxic porewater …"

L137-138: Did your killed controls have any sediment in them? This sentence reads as if only the bottles were autoclaved and 13C-methane was added to them.

Yes, the killed controls contained slurry that was autoclaved twice. This was clarified in the text by switching the word "bottle" to "slurry": "The "killed" control slurries in each experiment…"

L142-144: Id rather like to see how AQDS was prepared for your experiments and not what it has been shown to do; which was already introduced in the intro.

The sentence was changed as suggested to: "The synthetic analog for humic substances, i.e., 9,10-anthraquinone-2,6-disulfonate (AQDS), was dissolved in DDW (detailed in the supplementary information) and added to the bottles of experiment no. 6 until a final concentration of 5 mM was achieved in each bottle."

L139-154: This paragraph would read better if you included which experiment number or set corresponds to which electron acceptor or shuttler like you do with the inhibitors in the end of the paragraph.

The experiment numbers were added as suggested.

L151-154: Are experiment numbers the same as "sets" in L130? Please be consistent with your identifiers.

Yes. The text was corrected: "This study presents ten two-stage incubation experiments (experiment numbers 1-10) with different treatments…"

L151: Is there a reason why molybdate was only added to the magnetite samples?

Molybdate was added to one of the natural slurries, to one of the magnetite treatments, and to the slurries with the amorphous iron. This detail was clarified in the revised version.

L155: A sentence after this one describing why duplicate or triplicates were made would be useful.

We attempted to use triplicates whenever it was possible. In some cases, due to limitations associated with the slurries, we had to use duplicates. This was explained in the revised version: "All live treatments were set up in duplicate or triplicate, depending on the amount of the pre-incubated slurry aimed for each experiment and the results are presented as the average with an error bar."

L159: Please provide which lake and why did you pick a different lake to get humic substances.

A thermokarst lake near Fairbanks, Alaska. Unfortunately, natural humic substances were not available from Lake Kinneret, and thus we used humic substances that were available for us from another freshwater environment with profiles that exhibited similar methanogenic zone with iron reduction. These details were added to the text.

L161: So far I have not seen how your geochemistry (iron (II) and methane) are measured. I assume there is a section for that but perhaps add a parenthetical to indicate that.

Yes, there is a section for the geochemistry measurements (2.3.1) that is now indicated as suggested with the respective heading in the Analytical methods (2.3).

L167-168: When were these sediments collected and by what method. Were they collected by gravity corers like experiment A? Please elaborate. If all sediments for all experiments were collected between 2017 and 2019 by gravity corers etc… then that should be stated before all the experiments are described.

We agree, and therefore, a new subsection with the relevant sediment collection information was added as the referee suggested in a previous comment.

L169: Similarly, with the porewater. Is this porewater from the same methanogenic depth? Please add details or blanket the information before all the experiments are described.

Yes, the porewater was extracted from the same methanogenic depth as described in the new subsection. This was clarified in the text.

L171: What is the relative proportion of the 12C and 13C in the mixture?

We added 13 ml of $^{12}C$ and 2 ml of $^{13}C$ methane (total of 15 ml) to the bioreactors.

L174: What batch experiments?

The two-stage experiments and the fresh batch experiment. We clarified this in the revised version.

L175: Using an electrode not by electrode.

This was corrected.

L182-183: Are you saying the pre-incubation sediments were collected in 2013 instead of between 2017 and 2019? The timing of sample collection for the sediment samples for any of these experiments is not clear at all. I suggest that there should be a new section in the beginning of the methods describing when sampling occurred for which experimentation.

The sediments for all of the pre-incubations and for the bioreactors were collected between 2017-2019. Only the sediments for the fresh batch experiment were collected in 2013. We clarified this in the revised version in the new subsection, 2.2.1 General.

L185: Please provide the material of the experimental bottles.

These were glass bottles. This was added to the text.

L190-195: There is information in the figure caption that should be in the main part of the methods. Please move them accordingly.

The information was removed from the caption. It can be found in sub-section 2.2.2 (experiment type A set-up) of the revised version.

L198: I suggest this section needs to have subsections to disambiguate between geochemistry and molecular analysis. It is confusing to read about geochemical analysis and then jump to molecular analysis and then back to stable isotope analysis. Having sub headers would help the reader not only know when you are switching gears but also, for example, if someone wanted to find quickly a detail about the molecular analysis without reading the whole manuscript, they could easily find it using the sub headers.

We agree and sub-headers were added.

L206: Determined not "measured". You measured on a spectrophotometer.

The word measured was changed to "determined" as suggested.

L207-210: These two sentences do not read well. One could easily put them together for better flow.

The two sentences were combined as suggested: "A 100-µL headspace sample was taken for methane measurements with a gas-tight syringe and was analyzed by gas chromatograph (Focus GC, Thermo) equipped with a flame ionization detector (FID) and a packed column (Shincarbon ST) with a helium carrier gas (UHP) and a detection limit of 1 nmol $CH_4$."

L208-210: Please provide the GC model, column type and carrier gas.

The following details for the GC were added to the text: gas chromatograph (Focus GC, Thermo) equipped with flame ionization detector (FID) and packed column (Shincarbon ST) with a helium carrier gas (UHP).

L210-211: Ethylene was similarly determined to what? You mean that ethylene from acetylene reduction was measured by GC right?

Yes. This was clarified in the text.

L231: Not clear what is meant by duplicates a and b in the semi-bioreactor experiment.

This refers to the duplicate samples that were taken from the semi-bioreactor at that time point. They are called "a" and "b" in the supplementary coverage table. This was clarified in the text.

L235: Never good to start a sentence with a numerical. Consider starting with "A range of …" or spelling out nineteen.

The text was changed as suggested to: "Between 19 and 40 million…"

L245: Rate should be plural. Please check tenses throughout.

The word was corrected.

L262-263: Syntax error. This sentence reads as if the results from type A and B were compared to the batch slurry incubation presented by Bar-Or et al., and Elul et al.,. You mean that you are comparing YOUR type C experiments with Bar-Or et al., and Elul et al., type C experiments, correct?

Experiments type A (two stages) and B (fresh bioreactor) were performed during this study. We meant that these results were compared also to the fresh batch slurry incubations

presented by Bar-Or et al. and Elul et al. (type C), which were also set up by our group and co-authors. We clarified these details in the text.

L266: Which samples do you mean when you say natural non-killed slurries? Figure 2 and 3 do not have any identifiers called "Natural non-killed slurries". Do you mean all the slurries with different amendments? If so then I would not consider them natural. This is not clear and very confusing.

We meant all the non-killed treatments which were amended with $^{13}CH_4$. This was clarified in the text: "At the same time there was a conversion of $^{13}$C-methane to $^{13}$C-DIC in all the non-killed slurries amended with $^{13}$C-methane, indicating AOM (Figs. 3 and 4)."

L266: How "significant" is the AOM and relative to what? Did you do any statistics? I would be careful using this word.

The word was removed.

L266-268: Sentence does not read well, please restructure.

We rewrote the sentence: "The $\delta^{13}C_{DIC}$ values of the "methane-only" control slurries reached as high values as 743‰."

L270: I think that it is interesting that you have such a methanogenesis rate and Figure S3 should be moved into the main text. Also Figure S3 reports a rate in µM in the figure caption, mM in PW on the S3 y-axis and in nmol gr^-1 Day^-1. I suggest making it all one unit type as it is very confusing.

The figure was moved to the main text (Fig. 2) and we changed the units to nmol/gr dry sediment for concentrations and nmol/gr dry sediment*day for rates.

L271: Now it is a two-stage incubation and not long term please be consistent.

The long-term two-stage experiments are called "two-stage" experiments as a shorter version throughout most of the manuscript. This is clarified in subsection 2.2.2 Experiment A setup.

L275: What treatments? Do you mean replicates? Figure 2 suggests that they are replicates.

We meant to say "experiments" and not "treatments", we added hematite to the three two-stage experiments presented in Figure 2. This detail was corrected in the revised version.

L274-276: Sentence could be structured better.

The sentence was restructured as suggested: "The addition of hematite to three different experiments increased the $\delta^{13}C_{DIC}$ values over time to 694‰ (Fig. 3) similar to the behavior of the methane-only controls."

L282: Please move the Figure 3B citation to just after the describing the increase from the nontrite.

The citation was moved.

L286: AOM is not "of" the incubation it is "in"

The word "of" was changed as suggested.

L287: Was sulfate measured in this study? I do not recall it in the methods.

Sulfate was absent (BDL) in the natural sediments at the beginning of the experiments and thus was not measured later (it was also absent at our previous published experiments). However, we still tested its potential involvement as a short-lived intermediate with undetected concentrations by using a sulfate reduction inhibitor (molybdate) and did not find any evidence for that. We clarified this in the revised version (also in the discussion): "The involvement of sulfate in the AOM in the two-stage incubations was tested, even in the absence of detectable sulfate in the methanogenic sediments. This is as sulfate could theoretically still be a short living intermediate for the AOM process in an active cryptic sulfur cycle (Holmkvist et al., 2011)."

L289: I do not think that molybdate inhibits the sulfate reducing bacteria, it rather inhibits sulfate reduction by acting as an analog to sulfate and bind to APS enzyme. Please be careful with distinguishing organism with metabolism. This has also already been stated elsewhere, consider removing.

The repetitive text was removed.

L290-291: This sentence does not read well. Please restructure.

The sentence was restructured, and now reads: "This addition did not affect the increasing trend of $\delta^{13}C_{DIC}$ with time, and therefore, the AOM rates remained unchanged, similar to the observation in the fresh batch incubations (Bar-Or et al., 2017)."

L290-291: This is not so surprising since magnetite was added and iron reduction is not inhibited by molybdate and it outcompetes with sulfate reduction. If there is no sulfate/sulfide concentrations reported nor any rates, then does this have a place in this manuscript?

As mentioned above, sulfur species were tested for their possible involvement in the Fe-AOM in a cryptic cycle, even though the levels of dissolved sulfide and sulfate were negligible. This cryptic cycling occurs via oxidation of pyrite or FeS by iron oxides. We added molybdate to the methane-only treatment and to the magnetite treatment at the same time point. The addition of molybdate to the fresh batch experiment showed that sulfur species are not intermediates in the Fe-AOM process. Instead, competitive sulfur metabolism over the iron oxide occurs. In our two-stage experiment, we also observed that sulfate is not involved in the AOM. These details were clarified in the revised text.

L294-297: First time nitrate/nitrite measurements were mentioned. How did you measure it? This should rather be in the methods.

As for the sulfate, nitrate and nitrite were not detected in the natural methanic sediments over the years (Nüsslein et al., 2001; Sivan et al., 2011; Bar-Or et al., 2017; Elul et al., 2021). This was clarified, including the specific references.

L301-303: This sentence does not read well please reorganize.

The sentence was reorganized to read: "Following the addition of 0.5 mM of nitrite, we observed no increase in $\delta^{13}C_{DIC}$ values during the first 222 days (Fig. 4D), after which they increased from 34‰ to 54‰ by the end of the experiment."

L303-304: Is this AOM rate for nitrate or nitrite coupling?

This is the AOM rate for nitrite coupling, as explained in the revised text.

L320: Again BES is not an inhibitor of the methogen archaea it is an inhibitor of methanogenesis. You also need references.

This was corrected and the references were added.

L325: Methanogenesis not methanogens'

This was corrected.

L327: What does the (SN-#) in figure 2 mean? I think that part should be taken out since it has no meaning for the reader and could lead to some confusion. Also why aren't the rest of the 10 sets of the long term two stage results in the graph. Isn't the point to compare all three experiments in its entirety?

Each serial number (SN) refers to a specific experiment detailed in Table 1. This was clarified in the text. We use the SN to enable the reader to know to which experiment we are referring. We present only two treatments from each experiment presented in figure 3 (methane-only and hematite) and matching the experiments with serial numbers was the easiest way to identify which experiment we are referring to. The graph shows that there is no difference between the results of our two-stage methane-only and hematite addition

treatments, which is in contrast to what was observed previously in a fresh-batch experiment at Lake Kinneret (i.e. Bar-Or et al. 2017). For that, we could only use experiments that had methane-only and hematite addition treatments, and therefore, not all of the two-stage experiments are presented in Figure 2.

L345-348: Table 2 is a bit confusing. What is the serial number? How is there multiple treatments per serial number?

The SN refers to a specific two-stage experiment, as explained in the comment above. Each experiment consists of several treatments. We put the serial numbers of the experiments in table 2, so the reader can easily understand from which experiment the treatments were taken.

L375: Figure 5 is oddly placed after the results section of the molecular analysis. I suggest moving this as to not confuse the reader.

Figure 6 (previously Fig.5) was moved as suggested.

L399: Get rid of "our"

The word was removed.

L401-405: I think you should put the respective citation to the respective analysis in this sentence. For example, Figure 2 is not a representative of in-situ geochemical or microbial diversity profiles. Also this statement is rather generalized, are you saying this is happening everywhere or at Lake Kinneret?

The citations were moved to their respective analysis as suggested. The discussion there refers to what is known about Lake Kinneret sediments. This was clarified in the text: "The *in-situ* geochemical and microbial diversity profiles (Bar-Or et al., 2015) and the geochemical (Sivan et al., 2011; Bar-Or et al., 2017; Fig. 3) and metagenomic (Elul et al., 2021) analyses of batch incubations with fresh sediments provided strong support for the occurrence of Fe-AOM in sediments of the methanogenic zone below 20 cm."

L405-407: Are you talking about previous work with profiles or your results, not clear. If it is others please provide references.

We are talking indeed about previous work. The references were added.

L399-411: I am having a tough time distinguishing what is just a rereporting of findings from older publications and that of the interpretations of the present study. The point of the discussion is to interpret the meaning of results of the present study and use previous lit to support the argument.

We reduced the amount of introduction/background in that part of the discussion, keeping mainly what was done in this study. That paragraph now reads: "The *in-situ* geochemical and microbial diversity profiles (Bar-Or et al., 2015) and the geochemical (Sivan et al., 2011; Bar-Or et al., 2017; Fig. 3) and metagenomic (Elul et al., 2021) analyses of batch incubations with fresh sediments provided strong support for the occurrence of Fe-AOM in sediments of the methanogenic zone below 20 cm. Such profiles and alongside incubations showed an unexpected presence of aerobic bacterial methanotrophs together with anaerobic microorganisms, such as methanogens and iron reducers (Adler et al., 2011; Sivan et al., 2011; Bar-Or et al., 2015; Bar-Or et al., 2017; Elul et al., 2021). These findings suggested that both *mcr* gene-bearing archaea and aerobic bacterial methanotrophs mediate methane oxidation. In the current study, we have supportive evidence of considerable AOM in the long-term incubations, even after the two treatment stages and considering the low abundance of the microbial populations."

L416-417: It is weird to cite a different publication to describe your result. Instead of citing your figure along with a different reference, why not use the space to compare the findings in the present study with that of Bar-Or et al.,.

We added the comparison of the results here with that of Bar-Or et al.. 2017.

L416-422: Why is the B and C experiment being discussed here when the header suggests that the Long-term two stage incubations will be discussed? It seems like here you are comparing all three experiments rather than focusing on interpreting "Potential electron acceptors for AOM in the Long-term two-stage incubation experiments". This header is misleading as one of your experimental types (A) is called Long-term two-stage incubation.

We moved the title "Potential electron acceptors for AOM in the long-term two-stage incubation experiments" to the part of the discussion that, from there on, the text discusses the electron acceptors that were added to the long-term two-stage experiments.

L419-420: I think that the differences in the bioreactors are interesting, but I also think that it is not super surprising as slurries in replicate can act as independent communities, because they are removed from the original environment, amended and sit for such a long time.

We agree with the reviewer, this is indeed the main challenge in trying to mimic the original natural environment while working with cultures that are not pure. We, therefore, attempted to conduct several types of experiments that are close to natural systems and to learn from them about life in these sediments. We added a statement about this challenge in the revised version: "We assume that the observed difference in the bioreactors would have been more pronounced if methane concentrations had been higher, but it is still a significant finding. We also note that the difference between the bioreactors results may also be due to the fact that each bioreactor community developed separately."

L420-422: I also don't understand this statement. According to your data AOM was stimulated with hematite better than all the other amendments that were done.

That is correct, but since the data for the hematite addition are close to those of the methane-only treatment, we cannot conclusively state that the hematite addition stimulated the AOM.

L420: Again, did you do any statistics to suggest that it is significant?

The phrase "significant difference" was changed to "significant finding".

L425-428: Any lit to support that explanation?

It is merely a statement that if labeled (enriched) $^{13}$C methane is oxidized, $^{13}$C-DIC will be enriched, while alternatively, if natural organic carbon, which is isotopically light, is oxidized, $^{13}$C-DIC will be depleted.

L428-432: Run-on sentence, hard to follow and confusing please trim and reword.

The sentence was edited. Now reads: "Using mass-balance estimations in the methane-only and in the amorphous iron treatments and considering the DIC concentrations and $\delta^{13}C_{DIC}$ values of the methane-only treatments at the beginning of the experiment (6 mM and 60‰, respectively) and the values at the end (6.5 mM and 360‰, respectively), about 0.5 mM of the DIC was added by the AOM of methane with $\delta^{13}$C of ~4000‰."

L436-437: I am not sure your data suggest that adding iron decreases AOM activity directly. It is rather better to say iron is just being used for organoclastic iron reduction rather than to oxidize methane.

That text was revised, and now reads: "This means that adding amorphous iron to the system encouraged iron reduction that was coupled to the oxidation of organic compounds other than methane."

L437-441: You need references to back these statements up.

References were added.

L442-451: See comment to L290-291. I think this should be omitted or greatly reduced and stated in the beginning that you believe sulfur cycling does not play a role in your experiments.

Please see our response to comment L290-291. We believe that our addition of molybdate to the two-stage experimental system allowed us to observe whether the cryptic sulfur cycle is somehow involved in the AOM in the two-stage slurries. This is explained in the revised version.

L453-454: Ammonium oxidation with iron reduction needs a citation.

References were added

L455-459: I understand your results don't indicate nitrate supports AOM but I don't understand from your interpretations how nitrate delays AOM. Why would organoclastic denitrification outcompete AOM? There is no citation in the text that supports that claim. Denitrification would of course be outcompeting other organisms for organic material but AOM is only oxidizing methane, which you added in abundance. I would agree that it is strange to have no 13C DIC build up after adding nitrate but potentially, the addition of nitrate resulted in more organics to be oxidized a dilute the 13C signal. Furthermore, the text clearly stated ANME-2d was not found. So I think that it is more accurate to say that nitrate does not support AOM as an electron acceptor during your experiments because the known ANME group that uses nitrate was not present. Also figure 3 C and D do not show any trends with sediment amended with nitrite.

The text was adjusted as the referee suggested. It now reads: "Our results indicate that the addition of nitrate did not promote AOM, likely due to the absence of ANME-2d, which is known to use nitrate (Arshad et al., 2015; Haroon et al., 2013). In the case of nitrite, even low concentrations appeared to delay the increase in $\delta^{13}C_{DIC}$ values, suggesting that organoclastic denitrification outcompetes AOM, and despite the occurrence of Methylomirabilia, the role of nitrite-AOM is not prominent in the two-stage incubations (Figs. 4C, D). "
There was a typo in Figure 3, wherein panel D should present nitrite treatments ($NO_2$ and not $NO_3$). This was corrected in the revised version.

L461-463: This sentence is contradicting. You said they weren't directly measured but then they were high. It should rather say previous work has shown DOC concentrations to be high in Lake Kinneret…

This was corrected. It now reads: "Though humic substances were not measured directly in Lake Kinneret sediments, the DOC concentrations in the methanogenic-depth porewater were previously found to be high (~1.5 mM, Adler et al., 2011), suggesting that they may play a role in AOM."

L465-467: References

References were added.

L467: Cite your figure here. This is the one hit you got out of all the experiments and therefore, the crux. Please elaborate and compare to the Valenzuela et al.,

The figure was cited. We elaborated on this point and compared our results to those of Valenzuela et al. as suggested. The text now reads: "Yet, as was done by Valenzuela et al. (2017), the addition of natural humic substances did promote AOM, compared to the rest of the electron acceptors tested, and may thus support AOM (Fig. 4B)."

L479-482: Run-on sentence. Reads poorly.

The sentence was edited, now reads: "Methane oxidation in the pre-incubated Lake Kinneret sediments is likely mediated by either ANMEs or methanogens, as the addition of BES and acetylene immediately stopped the AOM (Fig. 6), similar to the results of the killed bottles and the BES treatment in the fresh batch experiment (Bar-Or et a., 2017)."

L487-497: Do the authors have any interpretations as to why the aerobic methanotrophs play a minor role? I think this paragraph could be much stronger a it is eluding to who is responsible for oxidizing methane. Consider structuring the paragraph to better nail who is overall responsible.

If there were micro levels of oxygen trapped in the natural sediments, they were likely used already during the first incubation period, but if not, they were exhausted by the beginning of the second stage of the incubation. The lack of oxygen damages the aerobic methanotroph population in the slurries, and therefore, we believe that they play only a minor role in the AOM observed during the two-stage experiments. The text now reads: "Using the isotopic compositions of specific lipids and metagenomics, we identified a considerable abundance of aerobic methanotrophs and methylotrophs in the fresh sediments, but not in the long-term slurries (Table 3, Fig. 7). In the natural sediments, micro levels (nano molar) of oxygen could be trapped in clays and slowly released to the porewater (Wang et al., 2018). However, if such micro levels of oxygen still existed during the time of the pre-incubation, they were probably already exhausted. Indeed, the results of our specific lipids and metagenomics analyses suggest that the aerobic methanotrophs lineages play only a minor role in the long-term slurries, probably due to complete depletion of the oxygen." The paragraph was restructured as suggested by the referee.

L520-521: The authors built this paragraph up nicely but if it is not back flux by methanogens then what could it be? This kind of ended abruptly.

We believe it to be AOM by ANME-1 or by methanogens by reverse methanogenesis with, to some extent, an external electron acceptor. This was clarified in the text, which now reads: "It, therefore, seems less likely that the observed DIC values in our study were sustained by methanogenesis back flux alone (without an external electron acceptor) than by active AOM, which, in this case, is probably performed by ANME-1 or by methanogens that perform reverse methanogenesis to some extent."

L529-531: This info could be useful in the discussion

We added this information to the discussion as suggested by the reviewer. Please see the response to comment L487-497.

---

## Author Response (AR3)

Point-by-point response to referee's comments (our answers in blue):

In the latest iteration of this manuscript by Vigderovich et al., the authors put considerable effort to the language, flow, and clarity through the whole document. The methods are much easier to follow and are much more reproduceable. The results are properly reported and discussed. However, I have only one minor comment which I will point to below. Otherwise, the remaining minor comments and edits are rather small and mostly cosmetic (see inline comments).

We thank the referee for the kind words and the thoughtful and thorough review. We addressed all the comments and revised the conclusions as suggested. We believe that the manuscript is now ready for publication.

Minor comment:

The results from the metagenomic and lipid analysis clearly show evidence of aerobic methane oxidizing bacteria. The discussion does clearly state that if aerobic methane oxidizing bacteria do play a role in turning over methane in these experiments is quite low. However, the conclusion reads as if they play a much bigger role in the turnover with methane which is hugely speculative. The problem I have with this portion of the manuscript is that aerobic methane oxidizing bacteria were not directly tested in any of the experimental setups. Oxygen concentrations were not determined, and the experiments were all set up anaerobically. Thus, the incubations were not setup to directly test for aerobic oxidation of methane activity. I still think though the metagenomic and lipid findings are a real bonus dataset and are very interesting and should be investigated further. However, I do not think the results presented in this paper warrant the statement in the conclusion that aerobic methane oxidizing bacteria play a role since it wasn't directly tested. I therefore, suggest that the paper would be much stronger by really highlighting the coupling of hematite to AOM by anaerobic archaea. Following this conclusion, provide examples of how aerobic oxidation of methane by the bacteria maybe involved in Lake Kinneret sediments, but the results presented here show a need for further direct testing of this potential.

We understand the points of this comment and the conclusions section was revised carefully accordingly.

Inline comments:

Overall, the manuscript reads much better. Below are some inline edits and comments I caught while reading. It is likely I did not catch all and I suggest the authors go back and thoroughly fix minor grammatical errors.

We addressed all the comments and fixed grammatical errors throughout the manuscript.

L54-55: There are other types of aerobic methane oxidizing bacteria. This sentence reads as if there is only 1 in existence. I suggest just adding in parentheticals type II and type X.

Type II and X were added in the parentheticals as suggested.

L52: get rid of "Thus"

The word was removed.

L79: Are you saying Fe-AOM is oxidizing methane in the methanogenic zone or Fe-AOM removes methane produced from methanogenic zone in the top 20 cm?

In Lake Kinneret sediments methane is produced from the top centimeter to at least 40 cm depth. Suggesting that this entire sediment column is the methanogenic zone. Methane concentrations peak at approximately 15 cm and then decrease. We consider the sediments below 15 cm to be the "deep methanogenic zone" where methane concentrations decrease due to Fe-AOM, which removes 10-15% of the produced methane at those depths (16-40 cm). We clarified the text: "iron coupled-AOM (Fe-AOM) removes 10-15% of the produced methane in the deeper part of the methanogenic zone (> 20 cm below the water-sediment interface)."

L88: Include "conditions" after "hypoxia"

The word "conditions" was added as suggested.

L94: Delete "are available"

The phrase was deleted.

L100: Add "of " before "two stages"

The word "of" was added.

L127: Please add how much 13C-methane you added.

The details were added. We refer to Table 1 for specific information regarding the amount of 13C-methane added.

L137-139: This would probably read better as two sentences.

The sentence was divided into two: "Semi-continuous bioreactor experiments in which sediments were collected up to three days before the experiment was set up (freshly sampled sediments). The sediment to porewater ratio was 1:4 and porewater was exchanged regularly."

L150: Cores were sliced at what cm intervals?

We used the bottom part (as a bulk) of each core (below 20 cm).

L151: Please provide details on the dedicated container. Also why not directly into falcon tubes?

We used a 5 L plastic container. We collect the bottom part of at least 10 cores. We transfer the sediment from the core straight to the container. The text was revised and clarified; it now reads: "The bottom part of the sediment cores (the deeper methanogenic zone, i.e., below 20 cm) was transferred, as a bulk, to a dedicated 5 L plastic container onboard."

L149-154: I am a bit confused here. You collected cores and extracted porewater on the same day. However, porewater was extracted in the lab while core collection and slicing happened on board. Were these just day long excursions and you were able to bring the sediments back to the lab for extraction? If so I suggest making this more clear that field sampling and laboratory processing happened on the same day of collection.

We understand the confusion, yes, these are day-long excursions. The cores were collected and the sediments for the porewater extraction were cut and transferred to a container on board in the morning. Then, in the lab, we transfer the sediments for the porewater extraction with cutoff syringes to 50 ml falcon tubes and centrifuge them to separate the porewater. All of this is done on the same day. This was clarified in the text: "Sediments from the deeper methanogenic zone (sediment depths > 20 cm) for the experiments were diluted with porewater from the methanogenic zone of parallel cores sampled on the same day. The bottom part of the sediment cores (below 20 cm) was transferred, as a bulk, to a dedicated 5 L plastic container while onboard. The cores and the container were brought back to the lab, and the porewater was extracted on the same day of sampling. In the lab, sediments were collected from the container with 20-ml cutoff syringes and moved to 50-ml falcon tubes."

L153: Syringe filtered or filter tower?

Syringe filtered. This was added to the text.

L155: Missing temperature units.

The units were added.

L155: Subsampled not subsamples

This was corrected.

L162: Was the N2 atmosphere maintained in a glove bag or continuous flushing? Please add.

Continuous flushing. This was added to the text.

L163: Please add how you homogenized the sediment.

We shook the sediments in the zipper bag. This was added to the text.

L163: Add the word "was" between "gr" and "transferred"

The word was added.

L258: Add spectrophotometer model

The spectrophotometer details were added.

L263-264: For ethylene determinations did you use the same column and carrier gas for these measurements? If so consider consolidating into the previous sentence with methane.

Yes, it was in the same GC-FID, this was added.

L326: How "significant" did you do statistics. I would consider removing.

The word was removed.

L350-352: Sentence does not read well.

The sentence was revised. It now reads: "It was quantified directly by adding Na-molybdate to the methane-only controls and to the magnetite amended treatments in the second stage long-term incubations (Fig. 4A)."

L352-354: The addition of what had no increasing effect to the 13C DIC pool? Molybdate or magnetite?

The addition of Na-molybdate. This was clarified in the revised version.

L479: How significant of a finding is this. Where are the statistics behind this claim? I would consider removing the word "significant" in the document when statistics was not done.

The word "significant" was replaced with the word "relevant".

L485: I think the flow of the manuscript would be better to have subsections to 4.2 for each of the electron acceptors (i.e. 4.2.1 various metal oxides as electron acceptors etc...).

We accept the referee's suggestion, and we added subsections for 4.2 section in the revised manuscript.

L603: You mention in the discussion that Aerobic methane oxidizing bacteria play a minor role in turning over methane. This was only drawn from lipid and fatty acid determinations and genomic evidence which doesn't necessarily mean they were active. Nor were any of these results connected to the 13C-DIC increases seen in the experiments. I therefore, suggest to adjust this sentence to say that the exact role of aerobic methane oxidizing bacteria in Lake Kinneret sediments needs further examination. Because as is, the text reads that they are more involved than what is reported in the data.

We agree with the referee and the whole conclusions section was revised. It now reads: "Previous results of geochemical and microbial profiles as well as incubations with fresh sediments from Lake Kinneret constitute evidence of the occurrence of Fe-AOM in the methanogenic zone. The process is performed by anaerobic archaeal methanogens and aerobic bacterial methanotrophs, which remove about 10-15% of the methane produced in the lake's sediment. In the current study, we found that after two incubation stages and intensive purging for a prolonged duration, AOM was still significant, consuming 3-8% of the

methane produced. However, the abundance of aerobic methanotrophs decreased and anaerobic archaea (ANME-1 or specific methanogens) appeared to be solely responsible for methane turnover. AOM could be a result of carbon back flux, as the methanogenic/AOM pathway is reversible, however, the high $\delta^{13}C_{DIC}$ signal points to a metabolic reaction. Terminal electron acceptors or electron shuttles stimulating Fe-AOM are either hematite and/or humic substances. The role of the aerobic methanotrophs of the order *Methylococcales*, which were found in the freshly collected sediment experiments, remains to be examined."